# Convergent Privacy Loss of Noisy-SGD without Convexity and Smoothness

**Eli Chien & Pan Li**
Department of Electrical and Computer Engineering
Georgia Institute of Technology
Georgia, U.S.A.
`{ichien6,panli}@gatech.edu`

## Abstract

We study the Differential Privacy (DP) guarantee of hidden-state Noisy-SGD algorithms over a bounded domain. Standard privacy analysis for Noisy-SGD assumes all internal states are revealed, which leads to a divergent Rényi DP bound with respect to the number of iterations. Ye & Shokri (2022) and Altschuler & Talwar (2022) proved convergent bounds for smooth (strongly) convex losses, and raise open questions about whether these assumptions can be relaxed. We provide positive answers by proving convergent Rényi DP bound for non-convex non-smooth losses, where we show that requiring losses to have Hölder continuous gradient is sufficient. We also provide a strictly better privacy bound compared to state-of-the-art results for smooth strongly convex losses. Our analysis relies on the improvement of shifted divergence analysis in multiple aspects, including forward Wasserstein distance tracking, identifying the optimal shifts allocation, and the Hölder reduction lemma. Our results further elucidate the benefit of hidden-state analysis for DP and its applicability.

## 1 Introduction

Noisy Stochastic Gradient Descent (Noisy-SGD), also known as DP-SGD (Abadi et al., 2016), is now the fundamental workhorse for privatizing machine learning models with the guarantee of differential privacy (DP) Dwork et al. (2006). The popularity of DP-SGD is due to its effectiveness and simplicity — it is nothing but SGD with per-sample gradient projected to a $\ell_2$ ball (also known as gradient clipping) and additive Gaussian noise. Despite the simplicity and ubiquity of the DP-SGD algorithm, we still do not have a holistic understanding regarding its privacy loss[1]. One such evidence is that standard privacy analysis based on composition theorem (Kairouz et al., 2015; Mironov, 2017) gives a divergent privacy loss with respect to the number of iterations. On the other hand, a naive but convergent privacy loss can be obtained by output perturbation when the domain is bounded. It shows that neither of these analyses provides a tight privacy bound, which worsens the privacy-utility trade-off by overestimating the amount of noise required for training with DP-SGD.

A natural question is open: *Is there a privacy loss bound that outperforms the Pareto frontier of standard composition and output perturbation?* Two recent seminal works (Altschuler & Talwar, 2022; Ye & Shokri, 2022) partially answer this question under various assumptions on the loss. They show that for smooth (strongly) convex losses over a bounded domain, the privacy loss of Noisy-SGD is convergent and lower than the output perturbation bound. Unfortunately, both works require losses to be smooth and (strongly) convex, which precludes the applicability of their results to more general problems beyond convexity and smoothness. Whether these restrictive assumptions can be relaxed is stated as an important open question in Ye & Shokri (2022); Altschuler & Talwar (2022). Specifically: *Can these restrictive assumptions be removed while still having a privacy loss bound that outperforms the Pareto frontier of standard composition and output perturbation?*

---

[1]We refer privacy loss to the DP parameters for the **last iterate** of Noisy-SGD.

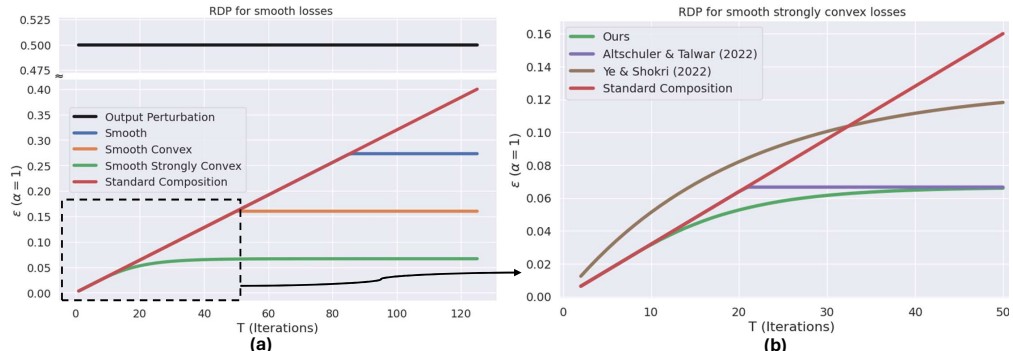

Figure 1: (a) Our RDP guarantees for smooth losses over the bounded domain, where the noise variance is the same for all lines. Orange and green lines indicate the cases where the loss is further assumed to be (strongly) convex. The output perturbation directly utilizes the Gaussian mechanism with sensitivity chosen to be the diameter of the bounded domain. (b) The detailed comparison of our privacy bound with Altschuler & Talwar (2022); Ye & Shokri (2022) for smooth strongly convex losses. The setting is the same as (a). We relegate the detail setting to Appendix A.13.

Table 1: Summary of the required assumptions of existing analysis. ✓ indicates the assumption is necessary and ☺ indicates one can optionally incorporate that assumption for a better privacy bound. †If we do not assume smoothness, we must have Hölder continuous gradient instead. ⋆If we assume strong convexity, the bounded domain assumption can be dropped.

|  | Strongly convex | Convex | Smooth | Bounded domain |
|---|---|---|---|---|
| Ye & Shokri (2022) | ✓ | ✓ | ✓ |  |
| Altschuler & Talwar (2022) | ☺ | ✓ | ✓ | ☺⋆ |
| Ours | ☺ | ☺ | ☺† | ☺⋆ |

## 1.1 Our Contributions and Analysis Overview

**Contributions.** We give a positive answer to the aforementioned open questions in this work. We show that the privacy loss of the Noisy-SGD algorithm is indeed convergent to a non-trivial value over a bounded domain even without convexity or smoothness assumptions. The least restrictive assumption we need is Hölder continuous (see Definition 2.5) gradients of order $\lambda \in (0, 1]$, which is a more general assumption compared to the loss being smooth. We say a function $f$ has Hölder continuous gradient with constant $L$ and order $\lambda$ if it satisfies $\|\nabla f(x) - \nabla f(y)\| \leq L\|x-y\|^\lambda$ for any $x, y$ in the domain. Specifically, when $\lambda = 1$ we recover the standard smoothness assumption. Note that there are non-smooth functions that still have Hölder continuous gradient. An iconic example is $f(x) = \text{sign}(x)\frac{3}{4}|x|^{4/3}$, where $f'(x) = |x|^{1/3}$. One can show that $f$ has Hölder continuous derivative of order $1/3$ with constant $2^{2/3}$ but not smooth (Appendix A.14). Even for a smooth function, the corresponding Hölder constant may be significantly smaller than the smooth constant for a smaller Hölder order $\lambda$, which potentially leads to a better privacy bound. See Figure 2 (a) for a neural network example. Note that this experiment is in the similar spirit of Zhang et al. (2020), where we do not assert that neural networks necessarily have Hölder continuous gradient in general. Instead, our goal is to demonstrate that generalizing hidden-state DP analysis to Hölder continuous gradient can be beneficial. Interestingly, our analysis also gives a strict improvement over existing privacy bounds (Altschuler & Talwar, 2022; Ye & Shokri, 2022) under the same set of assumptions with smoothness and strong convexity (Figure 1 (b)).

**Analysis.** We study the projected Noisy-SGD algorithm with per-sample gradient clipping, which is the projected version of the popular DP-SGD (Abadi et al., 2016). We defer all missing definitions in the preliminaries section 2. Let $\mathcal{D} = \{d_i\}_{i=1}^n \in \mathcal{X}^n$ be a training datasets of $n$ data points, where each data point $d_i$ associate with a loss function $\ell(\cdot; d_i)$ on a convex set $\mathcal{K}$ of diameter $D$. For any

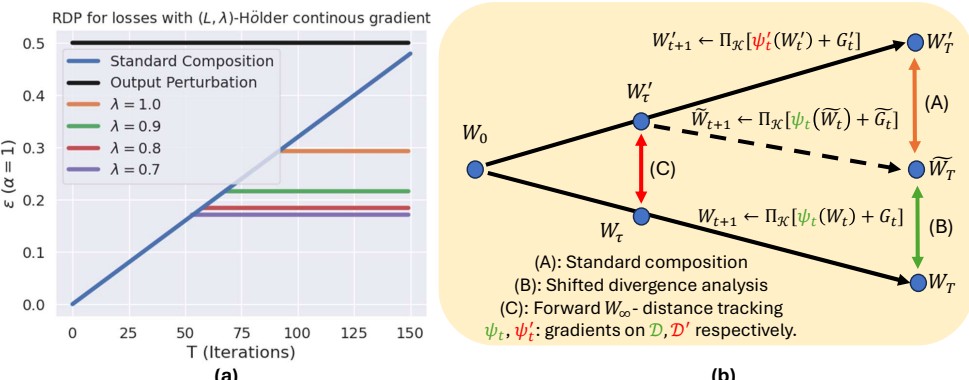

Figure 2: (a) Our RDP bound for non-smooth loss with $(L, \lambda)$-Hölder continuous gradient, where we empirically estimate the Hölder continuous constant $L$ of a 2 layer Multi-Layer Perceptron (MLP) for each $\lambda$. See Appendix A.13 for the detailed setting. (b) The illustration of the overall analysis. The decomposition of (A) + (B) parts is developed by Altschuler & Talwar (2022). It is done by constructing a coupling $(G_t, G_t')$, resulting a coupled process $\tilde{W}_t$. Part (A) is handled via standard composition or privacy amplification by subsampling in the mini-batch setting. Part (B) is handled by the shifted divergence analysis for smooth convex losses, which is also known as privacy amplification by iteration and will depend on the infinite Wasserstein distance $W_\infty(W_\tau, W_\tau')$. Altschuler & Talwar (2022) use the domain diameter $D$ as an upper bound. In contrast, we perform a careful forward $W_\infty$ distance tracking analysis (part (C)) to give a better bound, which provides a strict improvement to the final privacy loss bound. We further modify the analysis of part (B) so that it becomes applicable to even non-convex non-smooth losses with Hölder continuous gradients.

step size $\eta$, batch size $b$ and initialization $W_0 \in \mathcal{K}$, we iterate $T$ times the Noisy-SGD update

$$W_{t+1} = \Pi_{\mathcal{K}} \left[ W_t - \eta \frac{1}{b} \sum_{i \in \mathcal{B}_t} \Pi_{B_K} \left[ \nabla \ell_i(W_t) \right] + G_t \right], \tag{1}$$

where $\Pi_{\mathcal{K}}$ is the Euclidean projection to the closed convex set $\mathcal{K}$, $\Pi_{B_K}$ is the projection to the $\ell_2$ ball of radius $K$ (also known as gradient clipping), $G_t \sim N(0, \sigma^2 I)$ and $\mathcal{B}_t$ is the mini-batch of indices at time $t$. Our goal is to establish a worst-case upper bound of the Rényi divergence between distributions of the last iterate $W_T$ and $W_T'$ that are trained on any two adjacent datasets differing by one point. Prior work (Altschuler & Talwar, 2022) has shown that such Rényi divergence can be decomposed via coupling argument into two parts, see Figure 2 (b) for the illustration. Part (A) is the Rényi divergence between two processes with different gradient updates but the same start $W_\tau'$ for some intermediate time step $\tau$. This part can be handled via standard composition theorem and privacy amplification by subsampling. Part (B) is the Rényi divergence between two processes with the same gradient updates but different start $W_\tau, W_\tau'$. This part is handled by the shifted divergence analysis, which is originally known as privacy amplification by iteration (Feldman et al., 2018) and requires smoothness and convexity. Our main technical contributions are to improve the analysis of part (B), which allows us to not only relax the convexity and smoothness assumptions but also have a strictly tighter analysis for smooth strongly convex losses.

We first perform a careful forward Wasserstein distance tracking analysis (Lemma 3.5) until time step $\tau$ (part (C)). Compared to Altschuler & Talwar (2022) that utilize the domain diameter $D$ directly, our approach leads to a tighter bound for part (B) when further combined with the shifted divergence analysis. One design choice in the shifted divergence analysis is how we allocate the "shifts" for time step $t \in [\tau, T - 1]$. We identify the optimal shift allocations that not only lead to a superior privacy bound compared to the Pareto frontier of standard composition and output perturbation for smooth non-convex losses but also a strictly tighter privacy bound for smooth strongly convex losses. Our key intuition is that shifted divergence analysis provides us with a way to distribute the maximum discrepancy $W_\infty(W_\tau, W_\tau')$ to shifts in each time step $t \in [\tau, T - 1]$. This phenomenon does not rely on the convexity of losses, where the convexity only plays a role in the constraints that these shifts have to obey. To relax the smoothness assumption, the main challenge is that shifted divergence analysis strongly relies on the Lipschitz property of the gradient update (i.e., the Lipschitz reduction lemma 3.3) for establishing the proper constraints of shifts in each time

step. We alleviate this issue by proving the Hölder reduction lemma (Lemma 3.8) which allows us to work with the general Hölder continuous gradient condition.

## 2 PRELIMINARIES

**Notations.** We consider the empirical risk minimization problem, where the loss for weight $w$ on dataset $\mathcal{D}$ is $L(w; \mathcal{D}) = \frac{1}{n}\sum_{i=1}^{n} \ell(w; d_i)$. We denote $[n] := \{1, 2, \cdots, n\}$. We denote $f\sharp\mu$ the pushforward of a distribution $\mu$ under a function $f$. We denote $X_{i:j}$ as shorthand for the vector concatenating $X_i, \cdots, X_j$.

**(Rényi) differential privacy.** In this work we focus on replacement DP for simplicity, while extensions to notions of dataset adjacency are possible. We say two datasets $\mathcal{D}, \mathcal{D}'$ are adjacent (denoted as $\mathcal{D} \simeq \mathcal{D}'$) if one can be obtained from the other by replacing one data point arbitrarily. One may adopt the zero-out adjacency definition (Kairouz et al., 2021) for a smaller privacy loss as well. A popular way of deriving DP guarantees is to work with Rényi Differential Privacy (RDP) (Mironov, 2017), which relates to the Rényi divergence defined below.

**Definition 2.1** (Rényi divergence). For any $\alpha > 1$, the $\alpha$-Rényi divergence between two probability measures $\mu$ and $\nu$ is $D_\alpha(\mu||\nu) = \frac{1}{\alpha-1}\log\left(\int (\mu(x)/\nu(x))^\alpha \nu(x)dx\right)$ if $\mu \ll \nu$, and $\infty$ otherwise.

**Definition 2.2** (Rényi differential Privacy). A randomized algorithm $A$ satisfies $(\alpha, \varepsilon)$-RDP if $D_\alpha(\mathcal{A}(\mathcal{D})||\mathcal{A}(\mathcal{D}')) \leq \varepsilon$ for all possible $\mathcal{D} \simeq \mathcal{D}'$.

With a slight abuse of notation, for random variables $X \sim \mu$ and $Y \sim \nu$ we define $D_\alpha(X||Y) = D_\alpha(\mu||\nu)$. Rényi divergence has various nice properties, where we introduce the post-processing property and strong composition property as follows.

**Lemma 2.3** (Post-processing property). *For any $\alpha > 1$, any function $h$, and any probability distribution $\mu, \nu$, $D_\alpha(h\sharp\mu||h\sharp\nu) \leq D_\alpha(\mu||\nu)$.*

**Lemma 2.4** (Strong composition for Rényi divergence). *For any $\alpha > 1$ and any two sequences of random variables $X_1, \cdots, X_k$ and $Y_1, \cdots, Y_k$, $D_\alpha(X_{1:k}||Y_{1:k}) \leq \sum_{i=1}^{k} \sup_{x_{1:i-1}} D_\alpha(X_i|_{X_{1:i-1}=x_{1:i-1}}||Y_i|_{Y_{1:i-1}=x_{1:i-1}})$.*

Both of the above lemma are known to the literature and we refer interested readers to Altschuler & Talwar (2022) for a more thorough discussion.

**Potential assumptions on the loss $\ell$.** Throughout, the loss function corresponding to a data point $d_i$ or $d_i'$ is denoted by $\ell_i(\cdot) = \ell(\cdot; d_i)$ and $\ell_i'(\cdot) = \ell(\cdot; d_i')$ respectively. Let $\mathcal{K} \subset \mathbb{R}^d$ be an arbitrary closed convex set in which our model parameters $w$ lie.

**Definition 2.5** (Hölder continuity). For a function $f : \mathcal{K} \mapsto \mathbb{R}^d$, we say $f$ is $(L, \lambda)$-Hölder continuous if for all $w, w' \in \mathcal{K}$, $\|f(w) - f(w')\| \leq L\|w - w'\|^\lambda$, where $L \geq 0$ and $\lambda \in (0, 1]$.

Note that $(L, 1)$-Hölder continuous is equivalent to $L$-Lipschitz continuous. Thus a function has $(L, 1)$-Hölder continuous gradient is equivalent to $L$-smoothness when gradient exists. Hence, the $(L, \lambda)$-Hölder continuous gradient assumption is indeed more general. The other standard definitions such as Lipschitz, convexity, and smoothness are deferred to Appendix A.1.

**Shifted Rényi divergence analysis.** Our analysis generalizes the shifted Rényi divergence analysis developed by Feldman et al. (2018); Altschuler & Talwar (2022), where we merely need the Hölder continuous gradient instead of requiring the loss to be convex, smooth, and Lipschitz continuous. Below we introduce the definitions of infinite Wasserstein distance and shifted Rényi divergence which will play a central role in our proof.

**Definition 2.6** ($W_\infty$ distance). Let $\mu, \nu$ be probability distributions over $\mathbb{R}^d$. The infinite Wasserstein distance between $\mu, \nu$ is defined as $W_\infty(\mu, \nu) = \inf_{\gamma \in \Gamma(\mu,\nu)} \operatorname{ess\,sup}_{(X,Y)\sim\gamma} \|X - Y\|$, where $\Gamma(\mu, \nu)$ is the set of all possible coupling of $\mu, \nu$.

**Definition 2.7** (Shifted Rényi divergence). Let $\mu, \nu$ be probability distributions over $\mathbb{R}^d$. For any $z \geq 0$ and $\alpha > 1$, the shifted Rényi divergence is defined as $D_\alpha^{(z)}(\mu||\nu) = \inf_{\mu':W_\infty(\mu,\mu')\leq z} D_\alpha(\mu'||\nu)$.

## 3 OUR HIDDEN STATE DP-SGD PRIVACY LOSS ANALYSIS

Our goal is to show that the last iterate of the DP-SGD update (1) $W_T$ is $(\alpha, \varepsilon(\alpha))$-RDP and characterize the bound $\varepsilon(\alpha)$ under our weak assumptions. This is equivalent to bound the Rényi divergence of any two adjacent processes $W_T, W_T'$ defined by $\mathcal{D} \simeq \mathcal{D}'$ respectively in the worst case. The adjacent process of (1) with respect to the dataset $\mathcal{D}'$ is defined as follows

$$W_{t+1}' = \Pi_\mathcal{K}\left[W_t' - \eta\frac{1}{b}\sum_{i \in \mathcal{B}_t'}\Pi_{B_K}\left[\nabla \ell_i'(W_t')\right] + G_t'\right], \tag{2}$$

where the initializations are identical almost surely $W_0 = W_0'$.

Note that there are different possible mini-batch strategies adopted in practice such as without replacement subsampling, shuffled cyclic mini-batch, and full-batch cases. All of these are compatible with our analysis and we will study them separately. For simplicity, we start with the full batch setting $\mathcal{B}_t = [n]$ to demonstrate the key insight of our analysis.

### 3.1 FULL BATCH CASE WITH SMOOTH LOSSES

We start with introducing our first theorem, which gives a state-of-the-art privacy bound for smooth losses. Our theorem gives a lower privacy loss when more structural assumptions are presented, where we cover the case from strongly convex to non-convex losses in a unified manner.

**Theorem 3.1** (Privacy loss of Noisy-GD with smooth loss). *Assume $\ell$ is $L$-smooth. Then the Noisy-SGD update* (1) *with full batch setting is $(\alpha, \varepsilon(\alpha))$-RDP for $\alpha > 1$, where*

$$\varepsilon(\alpha) \leq \min_{\tau, \beta}\frac{\alpha}{2\sigma^2}\left(\sum_{t=\tau}^{T-1}\frac{(\frac{2K\eta}{n})^2}{\beta_t} + \frac{\min(\frac{2\eta K}{n}\sum_{t=0}^{\tau-1}c^t, 2\eta K\tau, D)^2}{\sum_{t=\tau}^{T-1}(1-\beta_t)c^{-2(t-\tau+1)}}\right), \tag{3}$$

$$s.t.\ \tau \in \{0, 1 \cdots, T-1\},\ \beta_t \in [0,1],\ \forall t \geq \tau,\ c = 1 + \eta L. \tag{4}$$

*If $\ell$ is also convex, $K$-Lipschitz and choose $\eta \leq 2/L$, we have $c = 1$. If $\ell$ is also $m$-strongly convex, $K$-Lipschitz and choose $\eta \leq 1/L$, we have $c = 1 - \eta m$.*

Theorem 3.1 indicates that after a burn-in period of iterations, there is no further privacy loss for both convex and non-convex cases as shown in Figure 1 (a). Our result degenerated to the results of Altschuler & Talwar (2022) for the convex case, and we show that a similar phenomenon also holds without convexity but requires a longer burn-in period of iterations with a larger final privacy loss. When the loss is $m$-strongly convex, our Theorem 3.1 provides a strict improvement over the result of Altschuler & Talwar (2022); Ye & Shokri (2022)[2]. See Figure 1 (b) for the illustration.

Below we provide a sketch of proof for Theorem 3.1 and the missing part can be found in Appendix A.4. The analysis presented in this section is the foundation of more general cases, such as non-smooth losses with Hölder continuous gradient and mini-batch settings. We start with introducing several technical lemmas that are crucial for our proof. The first two are the shift and Lipschitz reduction lemma, which establishes the relation of how an additive Gaussian noise and Lipschitz map affect the shifted Rényi divergence. They are the core of the shifted divergence analysis.

**Lemma 3.2** (Shift reduction lemma (Altschuler & Talwar, 2022; Feldman et al., 2018)). *For any probability distribution $\mu, \nu$ on $\mathbb{R}^d$, $\alpha > 1$ and $a, z \geq 0$, we have*

$$D_\alpha^{(z)}(\mu * N(0, \sigma^2 I)||\nu * N(0, \sigma^2 I)) \leq D_\alpha^{(z+a)}(\mu||\nu) + \frac{\alpha a^2}{2\sigma^2}. \tag{5}$$

**Lemma 3.3** (Lipschitz reduction lemma Altschuler & Talwar (2022)). *Assume the mapping $\phi$ is $c$-Lipschitz for $c \geq 0$. Then for any probability distribution $\mu, \nu$ on $\mathbb{R}^d$, $\alpha > 1$ and $z \geq 0$, we have*

$$D_\alpha^{(cz)}(\phi\sharp\mu||\phi\sharp\nu) \leq D_\alpha^{(z)}(\mu||\nu). \tag{6}$$

---

[2]In recent work concurrent to this paper, Altschuler et al. (2024) also proposed the improved analysis for strongly convex case that does not require a bounded domain and match our results asymptotically. Still, our analysis provides a strict improvement in the non-asymptotic regime.

On the other hand, one can show that the gradient update $x \leftarrow x - \eta \Pi_{B_K}[\nabla \ell(x)]$ is Lipschitz if the loss $\ell$ is at least smooth (Hardt et al., 2016; Altschuler & Talwar, 2023).

**Lemma 3.4** (Lipschitz constant of the gradient update map). *Let $\psi(x) = x - \eta \Pi_{B_K}[\nabla \ell(x)]$ be the gradient update map with loss $\ell$ and step size $\eta > 0$. If $\ell$ is $L$-smooth, $\psi$ is $1 + \eta L$ Lipschitz. If $\ell$ is further convex, $K$-Lipschitz and $\eta \le 2/L$, $\psi$ is $1$ Lipschitz. If $\ell$ is further $m$-strongly convex, $K$-Lipschitz and $\eta \le 1/L$, $\psi$ is $1 - \eta m$ Lipschitz.*

Finally, we introduce our forward Wasserstein distance tracking lemma, which is crucial to obtain strict improvement over the prior bounds for all number of iterations for smooth strongly convex losses. Notably, Chien et al. (2024b); Wei et al. (2024) have also adopted the similar idea of Wasserstein distance tracking analysis, but their analysis is for machine unlearning and DP-PageRank respectively, which are different from ours. See Appendix A.7 for a further discussion.

**Lemma 3.5** (Forward Wasserstein distance tracking for Lipschitz gradient update). *Consider the adjacent processes $W_t$, $W_t'$ defined in* (1) *and* (2) *respectively. Assume the gradient update map $\psi_t(x) = x - \frac{\eta}{n} \sum_{i \in [n]} \Pi_{B_K}[\nabla \ell_i(x)]$, $\psi_t'(x) = x - \frac{\eta}{n} \sum_{i \in [n]} \Pi_{B_K}[\nabla \ell_i'(x)]$ are $c$-Lipschitz for the full batch setting $\mathcal{B}_t = \mathcal{B}_t' = [n]$. Then we have*

$$W_\infty(W_t, W_t') \le \min(D_t, D), \ D_t = \min(cD_{t-1} + 2\eta K/n, D_{t-1} + 2\eta K), \ D_0 = 0. \quad (7)$$

Now we are ready to state the proof of sketch for Theorem 3.1.

*Proof.* We start with the argument of Altschuler & Talwar (2022), which constructs a specific coupling between the adjacent Noisy-SGD processes (1) and (2). To ease the notation, we denote $\psi_t, \psi_t'$ to be the gradient update map in (1) and (2) respectively and $\overset{d}{=}$ for equivalent in distribution.

$$W_{t+1} \overset{d}{=} \Pi_{\mathcal{K}}[\psi_t(W_t) + Y_t + Z_t], \ W_{t+1}' \overset{d}{=} \Pi_{\mathcal{K}}[\psi_t(W_t') + Y_t + Z_t'], \quad (8)$$

where $Y_t \sim N(0, (1 - \beta_t)\sigma^2 I)$, $Z_t \sim N(0, \beta_t \sigma^2 I)$ and $Z_t' \sim N(\psi_t'(W_t') - \psi_t(W_t'), \beta_t \sigma^2 I)$ for $\beta_t \in [0, 1]$. Notice that condition on $Z_t = Z_t'$, the two processes exhibit the **same** gradient update and additive noise. For any time step $0 \le \tau \le T - 1$ to be chosen later, we adopt this coupling for all $t \ge \tau$. Then by Lemma 2.4, we have the following decomposition of the privacy loss

$$D_\alpha(W_T || W_T') \le D_\alpha((W_T, Z_{\tau:T-1}) || (W_T', Z_{\tau:T-1}')) \quad (9)$$
$$\le D_\alpha(Z_{\tau:T-1} || Z_{\tau:T-1}') + \sup_{z_{\tau:T-1}} D_\alpha(W_T|_{Z_{\tau:T-1}=z_{\tau:T-1}} || W_T'|_{Z_{\tau:T-1}'=z_{\tau:T-1}}). \quad (10)$$

The first part can be handled by further applying the standard composition theorem (Lemma 2.4 for each time step), which leads to a bound $\sum_{t=\tau}^{T-1} \frac{\alpha}{2\beta_t \sigma^2} \left(\frac{2\eta K}{n}\right)^2$ and correspond to part (A) in Figure 2 (b). For the second term, we may iteratively apply Lemma 3.2, 3.3 to upper bound it with a shifted Rényi divergence term and additive Gaussian mechanism terms. Let us denote the shift for time step $t$ as $a_t \ge 0$, which will be determined later. With slight abuse of notation, we neglect the conditioning of $Z, Z'$ but keep in mind that the analysis below is under such conditioning.

$$D_\alpha(W_T || W_T') = D_\alpha^{(0)}(\Pi_{\mathcal{K}}[\psi_{T-1}(W_{T-1}) + Y_{T-1} + z_{T-1}] || \Pi_{\mathcal{K}}[\psi_{T-1}(W_{T-1}') + Y_{T-1} + z_{T-1}])$$

$$\overset{(a)}{\le} D_\alpha^{(0)}(\psi_{T-1}(W_{T-1}) + Y_{T-1} || \psi_{T-1}(W_{T-1}') + Y_{T-1})$$

$$\overset{(b)}{\le} D_\alpha^{(c^{-1}a_{T-1})}(W_{T-1} || W_{T-1}') + \frac{\alpha a_{T-1}^2}{2(1 - \beta_{T-1})\sigma^2}$$

$$\le D_\alpha^{(\sum_{t=\tau}^{T-1} c^{-(t-\tau+1)}a_t)}(W_\tau || W_\tau') + \sum_{t=\tau}^{T-1} \frac{\alpha a_t^2}{2(1 - \beta_t)\sigma^2}.$$

where $(a)$ is due the projection and constant shift are 1-Lipschitz maps and thus by Lemma 3.3, $(b)$ is due to Lemma 3.2 and 3.3, where the Lipschitz constant $c$ can be obtained from Lemma 3.4 based on different assumptions on the losses. The last inequality is by applying the argument iteratively until time step $\tau$. Note that while the shifted divergence term is in general non-tractable, it is 0 when $W_\infty(W_\tau || W_\tau')$ is less then the accumulated shifts according to its definition 2.7. By enforcing

this constraint, we obtain a closed-form privacy loss bound to be optimized with respect to $\tau, \beta_t, a_t$, where the constraint can be characterized by forward Wasserstein distance tracking lemma 3.5.

$$\min_{\tau,\beta,a} \frac{\alpha}{2\sigma^2} \sum_{t=\tau}^{T-1} \left( \frac{1}{\beta_t}(\frac{2\eta K}{n})^2 + \frac{a_t^2}{(1-\beta_t)} \right), \; s.t. \; \sum_{t=\tau}^{T-1} c^{-(t-\tau+1)} a_t \geq \min(D_\tau, D), \quad (11)$$

where $\tau \in [0, T-1], \beta_t \in (0,1), a_t \geq 0$ and $D_t$ is defined in Lemma 3.5. Finally, observing this constrained optimization can be further simplified by characterizing the optimum solution as follows. By Cauchy-Schwartz inequality, we have

$$\left[ \sum_{t=\tau}^{T-1} \frac{a_t^2}{(1-\beta_t)} \right] \cdot \left[ \sum_{t=\tau}^{T-1} (1-\beta_t)(c^{-2(t-\tau+1)}) \right] \geq (\sum_{t=\tau}^{T-1} a_t c^{-(t-\tau+1)})^2, \quad (12)$$

where the equality holds if the ratio $\frac{a_t}{(1-\beta_t)c^{-(t-\tau+1)}}$ is the same for all $\tau \leq t \leq T-1$. Apparently, this is attainable by choosing $a_t$ properly according to $\beta_t, c^{-(t-\tau+1)}$. Note that $(1-\beta_t)$ and $c^{-(t-\tau+1)}$ are all non-negative so that this characterization of $a_t$ still matches its non-negative constraint. As a result, the optimization problem above can be further simplified as stated in Theorem 3.1, where we complete the proof. □

It is worth noting the interpretation of Theorem 3.1, which is the root of our key intuition stated in Section 1.1. Under the optimum choice of $a_t$ determined by (12), we know that $a_t \propto (1 - \beta_t)c^{-(t-\tau+1)}$ where the left-hand side is closely related to the denominator of the second term $(1-\beta_t)c^{-2(t-\tau+1)}$ in the privacy bound of Theorem 3.1. As a result, the second term in the privacy bound of Theorem 3.1 can be interpreted as how we distribute the maximum discrepancy $\min(\frac{2\eta K}{n} \sum_{t=0}^{\tau-1} c^t, 2\eta K \tau, D)$ to the shifts $a_t$ each time step $t \in [\tau, T-1]$ along with weights $c^{-(t-\tau+1)}$. Even if the losses are non-convex so that $c > 1$, it is still possible to distribute the maximum discrepancy to more than one shift and thus a bound better than output perturbation is possible. We emphasize the discussion so far still relies on the smoothness assumption. Relaxing the smoothness assumption requires further non-trivial analysis as presented in the following section.

*Remark* 3.6. The focus of our work is on the privacy upper bound. The utility analysis for smooth strongly convex case can be found and adopted from Chourasia et al. (2021). In the meanwhile, since our work degenerate to the bound of Altschuler & Talwar (2022) for the smooth convex case, their lower bound analysis, especially the worst-case construction, applies to our scenario as well. We leave a full discussion on these aspects as future works.

## 3.2 NON-SMOOTH LOSSES WITH HÖLDER CONTINUOUS GRADIENT

Prior hidden-state Noisy-SGD privacy analysis requires the loss to be smooth (Altschuler & Talwar, 2022; Ye & Shokri, 2022), which restricts their application to non-smooth problems. Our Theorem 3.7 shows that Hölder continuous gradient is sufficient to enable the similar phenomenon of convergent privacy loss after a burn-in period. See Figure 2 (a) for an illustration.

**Theorem 3.7** (Privacy loss of Noisy-GD with Hölder continuous gradient). *Assume $\nabla \ell$ are $(L, \lambda)$-Hölder continuous with $\lambda \in (0, 1]$. Let $g(x) = x + \eta L x^\lambda$ with domain and range being $\mathbb{R}_{\geq 0}$ and $h = g^{-1}$. Then the Noisy-SGD update* (1) *with full batch setting is $(\alpha, \varepsilon(\alpha))$-RDP for $\alpha > 1$, where*

$$\varepsilon(\alpha) \leq \min_{\tau,\beta,a} \frac{\alpha}{2\sigma^2} \sum_{t=\tau}^{T-1} \left( \frac{1}{\beta_t}(\frac{2\eta K}{n})^2 + \frac{a_t^2}{1-\beta_t} \right), \quad (13)$$

$$s.t. \; \tau \in \{0, 1 \cdots, T-1\}, \; \beta_t \in [0,1], \; a_t \geq 0 \; \forall t \geq \tau, \; A_\tau \geq D_\tau, \quad (14)$$

$$A_T = D_0 = 0, \; A_{t-1} = h(A_t + a_{t-1}), \; D_t = \min(g(D_{t-1}) + 2\eta K/n, D_{t-1} + 2\eta K, D). \quad (15)$$

Below we provide the key lemmas that enable the analysis in Section 3.1 to be generalized to non-smooth losses with Hölder continuous gradient. Note that when the loss is non-smooth, the gradient update map $\psi$ can no longer be guaranteed to be Lipschitz, and thus Lemma 3.3 cannot be utilized. We prove the following key lemma which allows us to work without smoothness assumption.

**Lemma 3.8** (Hölder reduction lemma). *Assume the map $\phi$ is $(L, \lambda)$-Hölder continuous for $L \geq 0$ and $\lambda \in (0, 1]$. Then for any probability distribution $\mu, \nu$ over $\mathbb{R}^d$, $\alpha > 1$ and $z \geq 0$,*

$$D_\alpha^{(z)}((I + \phi)\sharp\mu||(I + \phi)\sharp\nu) \leq D_\alpha^{(h(z))}(\mu||\nu), \tag{16}$$

*where $h(z)$ is the solution map of the equation $x + Lx^\lambda = z$ for $x \geq 0$ and $I$ is the identity map.*

Note that the gradient update map $\psi$ can indeed be written as $I + \phi$, where $I$ is the identity map and $\phi$ is the gradient term. We further prove that the map $h$ is "well-behaved".

**Lemma 3.9** (Properties of $h$). *For any $z \geq 0$, let $h(z)$ be the solution map of the equation $x + Lx^\lambda = z$ for $x \geq 0$. For any $L \geq 0$ and $\lambda \in (0, 1]$, we have 1) $h : \mathbb{R}_{\geq 0} \mapsto \mathbb{R}_{\geq 0}$, 2) $h$ is strictly monotonic increasing and continuous, 3) $h$ is bijective, 4) when $\lambda = 1/2$, we have the close-form characterization $h(z) = \left( \frac{-L+\sqrt{L^2+4z}}{2} \right)^2$.*

Finally, we perform the forward $W_\infty$ distance tracking in this case.

**Lemma 3.10** (Forward Wasserstein distance tracking). *Consider the adjacent processes $W_t, W_t'$ defined in (1) and (2) respectively. Assume that $\nabla\ell_i(x), \nabla\ell_i'(x)$ are $(L, \lambda)$-Hölder continuous for all $i \in [n]$. Denote $g(x; L) = x + Lx^\lambda$, then we have*

$$W_\infty(W_t, W_t') \leq \min(D_t, D), \ D_t = \min(g(D_{t-1}; \eta L) + 2\eta K/n, D_{t-1} + 2\eta K), \ D_0 = 0. \tag{17}$$

The proof of Theorem 3.7 follows by a similar analysis in Section 3.1, but replacing the Lipschitz reduction lemma with Lemma 3.8 and Lemma 3.5 with Lemma 3.10. The full proof can be found in Appendix A.2 by specializing the minibatch results to full batch $[n]$ setting.

### 3.3 MINI-BATCH CASES

Our analysis naturally supports two different popular mini-batch settings: subsampling without replacement and shuffled cyclic mini-batch. While the privacy analysis for subsampling without replacement is more elegant, shuffled cyclic mini-batch can be implemented more efficiently and is closer to the standard mini-batch construction. We recommend interested readers to the recent excellent study on their pros and cons when deploying them in practice (Chua et al., 2024).

**Subsampling without replacement.** To express the result, we first introduce the Sampled Gaussian Mechanism (Mironov et al., 2019).

**Definition 3.11** (Rényi divergence of Sampled Gaussian Mechanism). *For any $\alpha > 1$, mixing probability $q \in (0, 1)$ and noise parameter $\sigma > 0$, define*

$$S_\alpha(q, \sigma) = D_\alpha(N(0, \sigma^2)||(1 - q)N(0, \sigma^2) + qN(1, \sigma^2)). \tag{18}$$

Note that $S_\alpha$ can be computed in practice with a numerically stable procedure for precise computation Mironov (2017). Now we are ready to state the result for subsampling without replacement.

**Theorem 3.12.** *Assume $\nabla\ell_i, \nabla\ell_i'$ are $(L, \lambda)$-Hölder continuous for $L \geq 0$ and $\lambda \in (0, 1]$. Let $h$ be the solution map defined in Lemma 3.8 with constant $\eta L, \lambda$. Then the Noisy-SGD update (1) under without replacement sampling mini-batches of size $b$ is $(\alpha, \varepsilon(\alpha))$-RDP for $\alpha > 1$, where*

$$\varepsilon(\alpha, \mathcal{B}) = \min_{\tau, \beta, a} \sum_{t=\tau}^{T-1} \left( S_\alpha(\frac{b}{n}, \frac{\beta_t \sigma b}{2\eta K}) + \frac{\alpha a_t^2}{2\sigma^2(1 - \beta_t)} \right), \tag{19}$$

$$s.t. \ \tau \in \{0, 1 \cdots, T - 1\}, \ \beta_t \in [0, 1], \ a_t \geq 0 \ \forall t \geq \tau, \ A_\tau \geq D_\tau, \ A_T = D_0 = 0, \tag{20}$$

$$A_{t-1} = h(A_t + a_{t-1}), \ D_t = \min(g(D_{t-1}; \eta L(b - 1)/b) + 2\eta K/b, D_{t-1} + 2\eta K, D), \tag{21}$$

*where $g(x; L) = x + Lx^\lambda$ and $S_\alpha(q, \sigma)$ is defined in Definition 3.11.*

The proof is similar to the proof of Theorem 3.7, except that we adopt the Sampled Gaussian Mechanism $S_\alpha$ to bound the first term (i.e., part (A) in Figure 2 (b)), which can be found at Appendix A.3. One can further specialize Theorem 3.12 if more structure assumptions are given, which leads to the subsampling without replacement version of Theorem 3.1. We leave it as an exercise for readers.

**Shuffled cyclic mini-batch.** The analysis for shuffled cyclic mini-batch is more tedious, but the main idea still follows Section 3.1. Due to the space limit, we defer the results to Appendix A.2.

## 4 RELATED WORKS

There are three types of privacy analysis for hidden-state Noisy-SGD, which are related to either Langevin dynamic, contraction of hockey-stick divergence, and shifted divergence. Chourasia et al. (2021) is the first that leverages Langevin dynamic analysis for deriving convergent privacy bound in the full batch setting. Ye & Shokri (2022) extend and refine their analysis to the mini-batch setting, yet both of which require smooth strongly convex losses. Chien et al. (2024a) also leverage the Langevin dynamic analysis but for machine unlearning problem. In the meanwhile, Asoodeh & Diaz (2023) utilize the bounded domain property with the contraction analysis of the hockey-stick divergence. Unfortunately, their bound heavily relies on the privacy amplification by subsampling effect, where their result degenerates to output perturbation in the full batch setting and is different from our approach. Nevertheless, there are other benefits regarding this analysis, such as supporting Poisson subsampling and without the need of RDP-DP conversion. It is interesting to see if the one can achieve the best of the both worlds. Finally, Altschuler & Talwar (2022) developed the analysis that combines privacy amplification by subsampling and iteration (Feldman et al., 2018) in a clever way, which leads to the state-of-the-art privacy bound for Noisy-SGD under smoothness and convexity assumption. Altschuler & Talwar (2023); Chien et al. (2024b) adopted a similar analysis for studying the mixing time of Noisy-SGD and its unlearning guarantee respectively. We improve their results with not only a generalization to non-smooth non-convex losses with Hölder continuous gradients, but also a tighter bound for the smooth strongly convex losses.

After the submission of our work, we notice that there are two recent works on hidden-state DP for Noisy-SGD. Annamalai (2024) shows that not all non-convex losses can be benefitted from hidden-state in privacy. This does not contradict our results, as we show that some "continuity" in gradient (i.e., Hölder continuous gradient) is needed for the reduction lemma type of result. Concurrently, Kong & Ribero (2024) study the hidden-state DP analysis of Noisy-SGD but with different condition to control the degree of convexity and do not consider the case of Hölder continuous gradient. Under the same set of assumptions on smoothness and convexity, our bound is provably tighter. See Appendix A.15 for a more detail comparison.

## 5 CONCLUSIONS AND OPEN PROBLEMS

We provide an affirmative answer to the open question raised in Altschuler & Talwar (2022); Ye & Shokri (2022): hidden-state Noisy-SGD indeed has non-trivial convergent privacy loss over the bounded domain even when the convexity and smoothness assumptions are relaxed. Our analysis shows that requiring losses to have Hölder continuous gradient is sufficient for convergent privacy loss that is better than the Pareto frontier of standard composition analysis and output perturbation. To the best of our knowledge, this is the least restrictive assumption in the literature. We further provide the superior privacy bound for smooth strongly convex losses compared to prior works.

While our results make a step forward toward the ultimate goal of providing a better privacy-utility trade-off of Noisy-SGD (or equivalently, DP-SGD) for training deep neural networks, several gaps remain. We discuss some future directions that can further progress in this direction.

**Gradient clipping.** While our Theorem 3.7 and 3.1 directly rely on gradient clipping in non-convex cases, it additionally requires losses to have bounded gradient norm "continuously". For simplicity, we directly require losses to be Lipschitz to satisfy this condition. While the gradient clipping operation can also ensure the bounded gradient norm, it inevitably induces discontinuous derivative of gradients on a set with negligible measure. The same open question is raised in Altschuler & Talwar (2022). The key question is whether we can prove the gradient update map $\psi(x) = x - \eta\Pi_{B_K}[\nabla f(x)]$ to be $c \le 1$ Lipschitz when the loss $f$ is known to be (strongly) convex and smooth. We conjecture this analysis can be done but potentially require additional assumptions on $f$. The concurrent work (Kong & Ribero, 2024) prove that the clipped gradient update map without Lipschitz loss assumption is indeed a Lipschitz map. It is interesting to further combine their result with our analysis as a future work.

**Better practical privacy accounting.** For ease of analysis, current hidden-state Noisy-SGD literature treats the model weight as a whole real vector as well as the assumptions of loss corresponding to it. This inevitably makes the corresponding Lipshitz, smooth, or Hölder constants large for neural networks even if they are naturally or modified to satisfy these assumptions. A recent interesting

work (Béthune et al., 2024) proposes to directly enforce layer-wise Lipschitzness of neural networks, so that gradient clipping can be dropped to reduce the time and space complexity of DP-SGD. They show that a per-layer analysis provides a better privacy-utility trade-off compared to requiring the Lipschitzness of the neural network as a whole. We conjecture a similar idea is necessary to improve the practicality of hidden-state Noisy-SGD analysis.

**Non-uniform Hölder continuous gradient.** Another aspect of improving the practicality of our work is to consider the non-uniform Hölder continuous gradient. Indeed, whenever the parameter difference $\|x - y\| \leq 1$, a smaller Hölder constant can be obtained for $\lambda < 1$. One interesting idea is to introduce the non-uniform (local) Hölder continuous gradient assumption for neural networks depending on the parameter difference. We conjecture that this is closer to the actual behavior of neural networks and potentially gives smaller constants and thus final privacy loss bound. This also requires additional investigation into what actual structural properties the neural networks have. We hope to see more collaboration between the theoretical and empirical ML community toward answering the last question.

**Uniform strict improvement over standard composition theorem.** Our current results show that there is no improvement before some burn-in period compared to the standard composition theorem for Noisy-SGD. A natural question to ask is whether a uniform strict improvement in the privacy-utility trade-off over the standard composition theorem can be made. Positive evidence to this question is the recent astonishing advances of DP-FTRL (Kairouz et al., 2021), which shows that a better privacy-utility trade-off can be achieved compared to standard composition theorem analysis if non-independent Gaussian noise is utilized. It is interesting to see if our analysis can be combined with DP-FTRL to further provide a better privacy-utility trade-off.

**Lower bound aspect of our analysis.** An interesting question is how tight is our derived privacy bound. We conjecture that our analysis is indeed relatively tight. The positive evidence is that our analysis degenerates to the result of Altschuler & Talwar (2022) for smooth convex losses, where they have constructed an order-wise matching lower bound on the privacy loss. We conjecture that a similar analysis can be conducted for our setting, which may elucidate the tightness of our analysis.

**Better privacy amplification by shuffling analysis.** While we have a privacy amplification by shuffling analysis from Ye & Shokri (2022) (see Corollary A.3), the resulting bound cannot be computed in a numerically stable way. Although Ye & Shokri (2022) propose a numerically stable approximation, it is unclear how such approximation affects the actual privacy guarantee. Ideally, we need a provable privacy upper bound for numerical approximate as in standard privacy accounting (Gopi et al., 2021) for rigorous DP guarantee. It is interesting to see if the privacy upper bound of the approximation method in Ye & Shokri (2022) can be established. A better privacy amplification by shuffling analysis is also an important open question as mentioned in Chua et al. (2024).

ACKNOWLEDGMENTS

We thank Wei-Ning Chen for the help in identifying the optimal shifts in our main theorems. We thank Rongzhe Wei for the helpful discussion on the idea of Wasserstein distance tracking. We thank Jason M. Altschuler for clarification regarding their papers. We thank Ziang Chen, Jiayuan Ye, Weiwei Kong, Shahab Asoodeh, reviewers and the area chair for the helpful discussion and comments. E.Chien and P. Li are supported by NSF awards CIF-2402816, PHY-2117997 and JPMC faculty award.

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

## A APPENDIX

### A.1 STANDARD DEFINITIONS

Let $f : \mathcal{K} \subseteq \mathbb{R}^d \mapsto \mathbb{R}$ be a mapping. We define smoothness, Lipschitzsness, and strong convexity as follows:

$$L\text{-smooth: } \forall\, x, y \in \mathcal{K},\ \|\nabla f(x) - \nabla f(y)\| \le L\|x - y\| \tag{22}$$

$$m\text{-strongly convex: } \forall\, x, y \in \mathcal{K},\ \langle x - y, \nabla f(x) - \nabla f(y)\rangle \ge m\|x - y\|^2 \tag{23}$$

$$M\text{-Lipschitz: } \forall\, x, y \in \mathcal{K},\ \|f(x) - f(y)\| \le M\|x - y\|. \tag{24}$$

Note that a function $f$ being $L$-smooth is equivalent to $f$ having $L$-Lipschitz continuous gradients. It is also equivalent to $f$ having $(L, 1)$-Hölder continuous gradient. Furthermore, we say $f$ is convex if it is 0-strongly convex.

### A.2 PRIVACY GUARANTEES FOR SHUFFLED CYCLIC MINI-BATCH

In this section, we assume for simplicity that $b$ divides $n$ so that $B = n/b$ is a positive integer[3]. Given the number of total iterations $T$, there will be $E = \lfloor T/B \rfloor + 1$ epochs so that $T = (E-1)B + \tilde{T}$. At the beginning of epoch $0 \le e < E$, we randomly partition the index set $[n]$ into $B$ mini-batches $\{\mathcal{B}_i^e\}_{i=0}^{B-1}$, where we denote $\mathcal{B}_t = \mathcal{B}_i^e$ if $t = eB + i$. Note that the mini-batch generation process is independent of the dataset $\mathcal{D}$ (as long as they have size $n$), the noise $G_t$, and the underlying model parameters $W_t$. In what follows, we will first derive the RDP guarantee for any fixed mini-batch sequence $\{\mathcal{B}_t\}_{t=0}^{T-1}$ and then improve the bound by taking into account the randomness of the mini-batches.

**Theorem A.1.** *Assume $\nabla \ell_i, \nabla \ell_i'$ are $(L, \lambda)$-Hölder continuous for $L \ge 0$ and $\lambda \in (0, 1]$. Let $h$ be the solution map defined in Lemma 3.8 with constant $\eta L, \lambda$. Given any mini-batch sequence $\mathcal{B} = \{\mathcal{B}_t\}_{t=0}^{T-1}$ from shuffled cyclic mini-batch strategy, define $\{t_e\}_{e=-1}^{E}$ be the set of time steps that we encounter $i^\star$ at epoch $e$, where $t_{-1} = 0$ and assume we will encounter $i^\star$ in the last epoch up to time step $T$. Then the DP-SGD update (1) with the given mini-batch sequence $\mathcal{B}$ is $(\alpha, \varepsilon(\alpha, \mathcal{B}))$-RDP for $\alpha > 1$, where*

$$\varepsilon(\alpha, \mathcal{B}) = \min_{\tau, \beta, a} \sum_{j=j_\tau}^{E} \frac{\alpha(\frac{2K\eta}{b})^2}{2\beta_{t_j}\sigma^2} + \sum_{t=\tau}^{T-1} \frac{\alpha a_t^2}{2\sigma^2(1-\beta_t)},\ j_\tau = j \text{ iff } \tau \in (t_{j-1}, t_j], \tag{25}$$

$$s.t.\ \tau \in \{0, 1 \cdots, T-1\},\ \beta_t = 0\ \forall t \notin \{t_e\}_{e=j_\tau}^{E},\ \beta_t \in [0, 1]\ \forall t \in \{t_e\}_{e=j_\tau}^{E},\ a_t \ge 0\ \forall t \ge \tau, \tag{26}$$

$$A_\tau \ge D_\tau,\ A_T = 0,\ A_{t-1} = h(A_t + a_{t-1}),\ D_t \text{ is defined in Lemma A.2.} \tag{27}$$

*If we do not encounter $i^\star$ in the last epoch up to time step $T$, replace $E$ by $E-1$.*

### A.2.1 THE ANALYSIS

We will need the following $W_\infty$ distance tracking lemma, which is crucial to contain privacy accounting based on the standard composition theorem (Mironov, 2017) as our special case. A similar idea is also used for proving unlearning guarantees for SGD unlearning (Chien et al., 2024b) and tight DP guarantees for the hidden-state PageRank algorithm (Wei et al., 2024).

**Lemma A.2** (Forward Wasserstein distance tracking)**.** *Consider the adjacent processes $W_t, W_t'$ defined in (1) and (2) respectively. Given any mini-batch sequences $\{\mathcal{B}_t\}$ and $\mathcal{B}_t = \mathcal{B}_t'$, denote $t_e$*

---

[3]When $b$ does not divide $n$, we can simply drop the last $n - \lfloor n/b \rfloor b$ points.

*be the time step that $i^\star \in \mathcal{B}_{t_e}$ at epoch $e$ so that $\lfloor t_e/B \rfloor = e$. Assume that $\nabla \ell(\cdot, d)$ is $(L, \lambda)$-Hölder continuous for any $d \in \mathcal{X}$. Then we have*

$$W_\infty(W_t, W_t') \leq \min(D_t, D), \tag{28}$$

$$D_t = \begin{cases} 0 & \text{if } t \leq t_0, \\ 2\eta K/b & \text{if } t = t_0 + 1, \\ \min(g(D_{t-1}; \eta L(b-1)/b) + 2\eta K/b, D_{t-1} + 2\eta K) & \text{if } t - 1 \in \{t_e\}_{e=1}^E, \\ \min(g(D_{t-1}; \eta L), D_{t-1} + 2\eta K) & \text{otherwise}, \end{cases} \tag{29}$$

*where $g(x; L) = x + Lx^\lambda$.*

Now we are ready to prove Theorem A.1.

*Proof.* The key idea of the shifted Rényi divergence analysis is constructing a coupling between $W_T, W_T'$ so that the analysis can be simplified (Altschuler & Talwar, 2022). According to our assumption, we have that $\mathcal{B}_t = \mathcal{B}_t'$ is some given fixed mini-batch sequence. Let us define the overall update map at time $t$ to be $\psi_t(W) = W - \eta \frac{1}{b} \sum_{i \in \mathcal{B}_t} \Pi_{B_K} [\nabla \ell_i(W)]$. Similarly, we can define $\psi_t'(W)$ to be the overall update map for dataset $\mathcal{D}'$. Then we construct a specific coupling between $G_t, G_t'$ as follows

$$W_{t+1} = \Pi_{\mathcal{K}} [\psi_t(W_t) + Y_t + Z_t], \quad W_{t+1}' = \Pi_{\mathcal{K}} \left[ \psi_t'(W_t') + Y_t + \tilde{Z}_t \right] \overset{d}{=} \Pi_{\mathcal{K}} [\psi_t(W_t') + Y_t + Z_t'], \tag{30}$$

where $Y_t \sim N(0, (1 - \beta_t)\sigma^2 I)$, $Z_t, \tilde{Z}_t \sim N(0, \beta_t \sigma^2 I)$ and $Z_t' \sim N(\psi_t'(W_t') - \psi_t(W_t'), \beta_t \sigma^2 I)$. It is worth noting that $\psi_t'(W_t') = \psi_t(W_t')$ for any $t$ such that $i^\star \notin \mathcal{B}_t$. Let us denote $t_e$ to be the time step such that $i^\star \in \mathcal{B}_t$ at epoch $e$. Then clearly $\psi_t, \psi_t'$ are only different at $\{t_e\}_{e=0}^E$.

Let us denote $t_{-1} = 0$. If we condition on $Z_t = Z_t'$ for all $t \in \{t_e\}_{e=j}^E$ for some $-1 \leq j \leq E$, then we know that $W_t, W_t'$ are two processes with identical update mapping $\psi_t = \psi_t'$ for any $t \geq t_j$. Thus, we can upper bound the Rényi divergence between $W_T, W_T'$ as follows: for any given $\tau \in \{t_e\}_{e=-1}^E$, let $j_\tau$ be the corresponding epoch index $e$ including $-1$. Let us denote $\hat{Z}_j = Z_{t_j}$ and $\hat{Z}_j' = Z_{t_j}'$, then we have

$$D_\alpha(W_T || W_T') \overset{(a)}{\leq} D_\alpha((W_T, \hat{Z}_{j_\tau:E}) || (W_T', \hat{Z}_{j_\tau:E}')) \tag{31}$$

$$\overset{(b)}{\leq} D_\alpha(\hat{Z}_{j_\tau:E} || \hat{Z}_{j_\tau:E}') + \sup_{z_{j_\tau:E}} D_\alpha(W_T|_{\hat{Z}_{j_\tau:E}=z_{j_\tau:E}} || W_T'|_{\hat{Z}_{j_\tau:E}'=z_{j_\tau:E}}), \tag{32}$$

where $(a)$ is due to post-processing property of Rényi divergence (Lemma 2.3) and $(b)$ is due to strong composition property (Lemma 2.4).

The first term corresponds to the standard composition theorem but starts only after time step $\tau$. Let us first analyze the quantity $m_{t_j} = \psi_{t_j}'(W_{t_j}') - \psi_{t_j}(W_{t_j}')$. Observe that for any $W$, we have

$$\|\psi_{t_j}'(W) - \psi_{t_j}(W)\| = \|\frac{\eta}{b} \sum_{i \in \mathcal{B}_{t_j}} \Pi_{B_K} [\nabla \ell_i(W)] - \Pi_{B_K} [\nabla \ell_i'(W)] \| \tag{33}$$

$$\overset{(a)}{=} \|\frac{\eta}{b} (\Pi_{B_K} [\nabla \ell_{i^\star}(W)] - \Pi_{B_K} [\nabla \ell_{i^\star}'(W)])\| \tag{34}$$

$$\leq \frac{\eta}{b} (\|\Pi_{B_K} [\nabla \ell_{i^\star}(W)]\| + \|\Pi_{B_K} [\nabla \ell_{i^\star}'(W)])\|) \tag{35}$$

$$\leq \frac{2K\eta}{b} \tag{36}$$

where $(a)$ is due to the fact at $t_j$ we have $\ell_i = \ell_i'$ except for $i = i^\star$ (i.e., the only data point that differs between $\mathcal{D}, \mathcal{D}'$). The rest is by triangle inequality and the projection operator $\Pi_{B_K}$. As a result, we know that $\|m_{t_j}\| \leq \frac{2K\eta}{b}$ for any $W_{t_j}$ and thus almost surely.

As a result, we can further bound the first term in (31) as follows

$$D_\alpha(\hat{Z}_{j_\tau:E} || \hat{Z}'_{j_\tau:E}) \tag{37}$$

$$\overset{(a)}{\leq} \sum_{j=j_\tau}^{E} \sup_{z_{j_\tau:j-1}} D_\alpha(\hat{Z}_j|_{\hat{Z}_{j_\tau:j-1}=z_{j_\tau:j-1}} || \hat{Z}'_j|_{\hat{Z}'_{j_\tau:j-1}=z_{j_\tau:j-1}}) \tag{38}$$

$$\overset{(b)}{=} \sum_{j=j_\tau}^{E} \sup_{z_{j_\tau:j-1}} D_\alpha(N(0, \beta_{t_j}\sigma^2 I) || N(m_{t_j}, \beta_{t_j}\sigma^2 I)) \tag{39}$$

$$\overset{(c)}{\leq} \sum_{j=j_\tau}^{E} \sup_{\|m_{t_j}\| \leq \frac{2K\eta}{b}} \frac{\alpha\|m_{t_j}\|^2}{2\beta_{t_j}\sigma^2} \tag{40}$$

$$\leq \sum_{j=j_\tau}^{E} \frac{\alpha(\frac{2K\eta}{b})^2}{2\beta_{t_j}\sigma^2}, \tag{41}$$

where $m_{t_j} = \psi'_{t_j}(W'_{t_j}) - \psi_{t_j}(W'_{t_j})$ condition on $\hat{Z}'_{j_\tau:j-1} = z_{j_\tau:j-1}$. $(a)$ is due to strong composition of Rényi divergence (Lemma 2.4), $(b)$ is by the definition of $\hat{Z}, \hat{Z}'$ (see above), $(c)$ is due to (33) and the close-form of Rényi divergence between two gaussian of the same variance but different mean Mironov (2017). Together we have successfully bound the first term.

For the second term in (31), note that it is now condition on $\hat{Z}_{j_\tau:E} = \hat{Z}'_{j_\tau:E} = z_{j_\tau:E}$ so that the two processes will have the same update iterations. For brevity, we omit the conditioning in the following derivation but all discussions are conditioning on $\hat{Z}_{j_\tau:E} = \hat{Z}'_{j_\tau:E} = z_{j_\tau:E}$ if not specified. For all $t \in \{t_e\}_{e=j_\tau}^E$, the two process becomes

$$W_{t_e+1} = \Pi_\mathcal{K}\left[\psi_{t_e}(W_{t_e}) + Y_{t_e} + z_e\right], \; W'_{t_e+1} = \Pi_\mathcal{K}\left[\psi_t(W'_{t_e}) + Y_{t_e} + z_e\right], \tag{42}$$

and for all other $t \notin \{t_e\}_{e=j_\tau}^E$, we already have

$$W_{t+1} = \Pi_\mathcal{K}\left[\psi_t(W_t) + Y_t + Z_t\right], \; W'_{t+1} = \Pi_\mathcal{K}\left[\psi_t(W'_t) + Y_t + \tilde{Z}_t\right]. \tag{43}$$

Note that we can simply choose the coupling for $\tilde{Z}_t$ to be $\tilde{Z}_t = Z_t$ for all $t \notin \{t_e\}_{e=j_\tau}^E$. Recall that by our definition of $Y, Z$, we know that $Y_t + Z_t = G_t \sim N(0, \sigma^2 I)$. Now we are ready to bound the second term in (31), which is done as follows. For all $t \in \{t_e\}_{e=j_\tau}^E$, we have for any $z_{j_\tau:E}$ and $z \geq 0$,

$$D_\alpha^{(z)}(W_{t_e+1} || W'_{t_e+1}) \tag{44}$$

$$= D_\alpha^{(z)}(\Pi_\mathcal{K}\left[\psi_{t_e}(W_{t_e}) + Y_{t_e} + z_e\right] || \Pi_\mathcal{K}\left[\psi_{t_e}(W'_{t_e}) + Y_{t_e} + z_e\right]) \tag{45}$$

$$\overset{(a)}{\leq} D_\alpha^{(z)}(\psi_{t_e}(W_{t_e}) + Y_{t_e} + z_e || \psi_t(W'_{t_e}) + Y_{t_e} + z_e) \tag{46}$$

$$\overset{(b)}{\leq} D_\alpha^{(z)}(\psi_{t_e}(W_{t_e}) + Y_{t_e} || \psi_t(W'_{t_e}) + Y_{t_e}) \tag{47}$$

$$\overset{(c)}{\leq} D_\alpha^{(z+a_{t_e})}(\psi_{t_e}(W_{t_e}) || \psi_t(W'_{t_e})) + \frac{\alpha a_{t_e}^2}{2(1-\beta_{t_e})\sigma^2} \tag{48}$$

$$\overset{(d)}{\leq} D_\alpha^{(h(z+a_{t_e}))}(W_{t_e} || W'_{t_e}) + \frac{\alpha a_{t_e}^2}{2(1-\beta_{t_e})\sigma^2}, \tag{49}$$

where $(a), (b)$ is due to the that both $\Pi_\mathcal{K}$ and constant translation are 1-Lipschitz. So the inequalities follow by the Lipschitz reduction lemma (Lemma 3.3). $(c)$ is due to shift reduction lemma (Lemma 3.2) for some $a_{t_e} \geq 0$ to be optimized later. $(d)$ is due to Hölder reduction lemma (Lemma 3.8), since $\psi = I + \phi$ where $\phi$ is the sum of mini-batch gradients. By our assumption that $\nabla\ell$ are $(L, \lambda)$-Hölder continuous (and thus $\eta\nabla\ell$ are $(\eta L, \lambda)$-Hölder continuous), indeed our Lemma 3.8 can be applied.

On the other hand, we can repeat a similar analysis for the case $t \notin \{t_e\}_{e=j_\tau}^E$, which leads to

$$D_\alpha^{(z)}(W_{t+1}||W'_{t+1}) \tag{50}$$

$$= D_\alpha^{(z)}(\Pi_\mathcal{K}[\psi_t(W_t) + G_t]\,||\Pi_\mathcal{K}[\psi_t(W'_t) + G_t]) \tag{51}$$

$$\overset{(a)}{\leq} D_\alpha^{(z)}(\psi_t(W_t) + G_t||\psi_t(W'_t) + G_t) \tag{52}$$

$$\overset{(b)}{\leq} D_\alpha^{(z+a_t)}(\psi_t(W_t)||\psi_t(W'_t)) + \frac{\alpha a_t^2}{2\sigma^2} \tag{53}$$

$$\overset{(c)}{\leq} D_\alpha^{(h(z+a_t))}(W_t||W'_t) + \frac{\alpha a_t^2}{2\sigma^2}, \tag{54}$$

where $(a)$ is due to the that both $\Pi_\mathcal{K}$ and constant translation are 1-Lipschitz. So the inequalities follow by the Lipschitz reduction lemma (Lemma 3.3). $(b)$ is due to shift reduction lemma (Lemma 3.2) for some $a_t \geq 0$ to be optimized later.

As a result, we can unroll $D_\alpha(W_T||W'_T)$ using the bounds (49) (54) above until time $t = \tau$, which leads to the following bounds. For brevity, we denote $1_t := 1\{t \in \{t_e\}_{e=j_\tau}^E\}$ as the indicator function of whether $t \in \{t_e\}_{e=j_\tau}^E$ or not. For any $z_{j_\tau:E}$,

$$D_\alpha(W_T||W'_T) = D_\alpha^{(0)}(W_T||W'_T) \overset{(a)}{=} D_\alpha^{(A_T)}(W_T||W'_T) \tag{55}$$

$$\leq D_\alpha^{(h(A_T+a_{T-1}))}(W_{T-1}||W'_{T-1}) + \frac{\alpha a_{T-1}^2}{2\sigma^2(1 - \beta_{T-1}1_{T-1})} \tag{56}$$

$$\overset{(b)}{\leq} D_\alpha^{(A_{T-1})}(W_{T-1}||W'_{T-1}) + \frac{\alpha a_{T-1}^2}{2\sigma^2(1 - \beta_{T-1}1_{T-1})} \tag{57}$$

$$\cdots \leq D_\alpha^{(A_\tau)}(W_\tau||W'_\tau) + \sum_{t=\tau}^{T-1} \frac{\alpha a_t^2}{2\sigma^2(1 - \beta_t 1_t)}, \tag{58}$$

where $(a), (b)$ is due to that we denote $A_T = 0$, $A_{t-1} = h(A_t + a_{t-1})$ for all $t \geq \tau + 1$. Finally, note that by Wasserstein forward tracking lemma (Lemma A.2), we know that $W_\infty(W_\tau, W'_\tau) \leq \min(D_\tau, D)$ (see Lemma A.2 for the definition of $D_t$). As a result, if $A_\tau \geq D_\tau$, then $D_\alpha^{(A_\tau)}(W_\tau||W'_\tau) = 0$. Thus by combining everything so far, we have the following optimization problem for the privacy loss

$$\min_{\tau,\beta,a} \sum_{j=j_\tau}^E \frac{\alpha(\frac{2K\eta}{b})^2}{2\beta_{t_j}\sigma^2} + \sum_{t=\tau}^{T-1} \frac{\alpha a_t^2}{2\sigma^2(1 - \beta_t)}, \tag{59}$$

$$s.t.\ \tau \in \{t_e\}_{e=-1}^E,\ \beta_t = 0\ \forall t \notin \{t_e\}_{e=j_\tau}^E,\ \beta_t \in [0,1]\ \forall t \in \{t_e\}_{e=j_\tau}^E,\ a_t \geq 0\ \forall t \geq \tau, \tag{60}$$

$$A_\tau \geq D_\tau,\ A_T = 0,\ A_{t-1} = h(A_t + a_{t-1}),\ D_t \text{ is defined in Lemma A.2.} \tag{61}$$

Together we complete the proof. $\qquad\square$

### A.2.2 IMPROVED BOUND BY SHUFFLING

Note that the privacy bound we derived in Theorem A.1 holds for any realization of shuffled cyclic minibatch sequences. As we can see, the bound indeed depends on the time step $\{t_e\}_{e=0}^E$ that we encounter $i^\star$ at each epoch $e$, which is inevitably controlled by the worst-case scenario when requiring a data-independent RDP bound. To alleviate the worst-case issue, it is critical to take the randomness of the shuffled cyclic minibatch sequence into account. The following corollary serves for this purpose

**Corollary A.3.** *Let $\varepsilon(\alpha, \mathcal{B})$ be the optimal privacy loss derived in Theorem A.1 give a cyclic minibatch sequence $\mathcal{B}$. Under the same assumption as in Theorem A.1, the DP-SGD update (1) is $(\alpha, \varepsilon(\alpha))$-RDP, where*

$$\varepsilon(\alpha) \leq \frac{1}{\alpha - 1} \log\left(\mathbb{E}_\mathcal{B} \exp\left((\alpha - 1)\varepsilon(\alpha, \mathcal{B})\right)\right). \tag{62}$$

### A.3 Privacy Guarantees for Without Replacement Subsampling Mini-batches

In this section, each mini-batch $\mathcal{B}_t$ is sampled independently and identically from $[n]$ for time step $t$ in the without replacement fashion of size $b$. That is, we randomly sample $\mathcal{B}_t$ out of the uniform random subset of size $b$ from $[n]$. This strategy of mini-batch sampling is known as without replacement subsampling. In what follows, we derive the RDP guarantee for DP-SGD update (1) under without replacement subsampled mini-batches.

#### A.3.1 The Analysis

Let us first introduce a technical lemma before we introduce our proof. Note that the original Sampled Gaussian Mechanism is defined in one dimension. Altschuler & Talwar (2022) extends this notion to a higher dimension by identifying the worst-case scenario therein. Alternatively, one can directly work with high-dimensional Sampled Gaussian Mechanism as in Altschuler et al. (2024).

**Lemma A.4** (Extrema of Sampled Gaussian Mechanism, Lemma 2.11 in Altschuler & Talwar (2022)). *For any $\alpha > 1$, $q \in (0,1)$ and the noise parameter $\sigma > 0$, dimension $d \in \mathbb{N}$ and radius $R > 0$,*

$$\sup_{\mu \in \mathcal{P}(B_R)} D_\alpha(N(0, \sigma^2 I_d) || (1-q)N(0, \sigma^2 I_d) + q(N(0, \sigma^2 I_d) * \mu)) = S_\alpha(q, \sigma/R), \quad (63)$$

*where $\mathcal{P}(B_R)$ denotes set of all Borel probability distributions over the $\ell_2$ ball of radius $R$ in $\mathbb{R}^d$.*

In practice, $S_\alpha(q, \sigma)$ is computed via numerical integral for the tightest possible privacy accounting. Nevertheless, to have a better understanding of the quantity, Lemma 2.12 in Altschuler & Talwar (2022) shows that it is upper bounded by $2\alpha q^2/\sigma^2$ for some regime of $(\alpha, \sigma, q)$. In what follows, we keep the notation of $S_\alpha(q, \sigma)$ but is useful to keep this simplified upper bound in mind.

Now we are ready to prove Theorem 3.12.

*Proof.* Following the same analysis, we have that

$$D_\alpha(W_T || W_T') \quad (64)$$

$$\leq D_\alpha(Z_{\tau:T-1} || Z_{\tau:T-1}') + \sup_{z_{\tau:T-1}} D_\alpha(W_T|_{Z_{\tau:T-1}=z_{\tau:T-1}} || W_T'|_{Z_{\tau:T-1}'=z_{\tau:T-1}}). \quad (65)$$

For the first term, we bound it as follows

$$D_\alpha(Z_{\tau:T-1} || Z_{\tau:T-1}') \quad (66)$$

$$\leq \sum_{t=\tau}^{T-1} \sup_{z_{\tau:t-1}} D_\alpha(Z_t|_{Z_{\tau:t-1}=z_{\tau:t-1}} || Z_t'|_{Z_{\tau:t-1}'=z_{\tau:t-1}}) \quad (67)$$

$$= \sum_{t=\tau}^{T-1} D_\alpha(N(0, \beta_t \sigma^2 I) || (1 - \frac{b}{n})N(0, \beta_t \sigma^2 I) + \frac{b}{n}N(m_t, \beta_t \sigma^2 I)) \quad (68)$$

$$\leq \sum_{t=\tau}^{T-1} S_\alpha(\frac{b}{n}, \frac{\beta_t \sigma}{2\eta K/b}), \quad (69)$$

where we use the fact that $\|m_t\| \leq 2\eta K/b$ almost surely, since each gradient term is bounded by $\eta K$ due to gradient clipping $\Pi_{B_K}$ and there is at most 1 out of $b$ term that corresponds to $i^\star$. Hence, one can apply Lemma A.4 for the last inequality. For the second term, the analysis follows similarly as in the proof of Theorem A.1. Hence we complete the proof. $\square$

### A.4 Proof of Theorem 3.1

*Proof.* We start with the argument of Altschuler & Talwar (2022), which constructs a specific coupling between the adjacent Noisy-SGD processes (1) and (2). To ease the notation, we denote $\psi_t, \psi_t'$ to be the gradient update map in (1) and (2) respectively. Then note that

$$W_{t+1} \overset{d}{=} \Pi_{\mathcal{K}}[\psi_t(W_t) + Y_t + Z_t], \ W_{t+1}' \overset{d}{=} \Pi_{\mathcal{K}}[\psi_t'(W_t') + Y_t + Z_t'], \quad (70)$$

where $Y_t \sim N(0, (1-\beta_t)\sigma^2 I)$, $Z_t \sim N(0, \beta_t\sigma^2 I)$ and $Z'_t \sim N(\psi'_t(W'_t) - \psi_t(W'_t), \beta_t\sigma^2 I)$ for $\beta_t \in [0, 1]$. Notice that condition on $Z_t = Z'_t$, the two processes exhibit the **same** gradient update and additive noise. For any time step $0 \le \tau \le T-1$ to be chosen later, we adopt this coupling for all $t \ge \tau$. Then by Lemma 2.4, we have the following decomposition of the privacy loss

$$D_\alpha(W_T||W'_T) \le D_\alpha((W_T, Z_{\tau:T-1})||(W'_T, Z'_{\tau:T-1})) \tag{71}$$

$$\le D_\alpha(Z_{\tau:T-1}||Z'_{\tau:T-1}) + \sup_{z_{\tau:T-1}} D_\alpha(W_T|_{Z_{\tau:T-1}=z_{\tau:T-1}}||W'_T|_{Z'_{\tau:T-1}=z_{\tau:T-1}}). \tag{72}$$

The first part can be handled by further applying Lemma 2.4 as follows.

$$D_\alpha(Z_{\tau:T-1}||Z'_{\tau:T-1}) \le \sum_{t=\tau}^{T-1} \sup_{z_{\tau:t-1}} D_\alpha(Z_t|_{Z_{\tau:t-1}-z_{\tau:t-1}}||Z'_t|_{Z'_{\tau:t-1}-z_{\tau:t-1}}) \tag{73}$$

$$= \sum_{t=\tau}^{T-1} \sup_{z_{\tau:t-1}} D_\alpha(N(0, \beta_t\sigma^2 I)||N(\psi'_t(W'_t) - \psi_t(W'_t), \beta_t\sigma^2 I)|_{Z'_{\tau:t-1}-z_{\tau:t-1}}) \tag{74}$$

$$\le \sum_{t=\tau}^{T-1} \frac{\alpha}{2\beta_t\sigma^2}\left(\frac{2\eta K}{n}\right)^2, \tag{75}$$

where the last step is due to the fact that $\|\psi'_t(W) - \psi_t(W)\| \le \frac{2\eta K}{n}$ for any $W$. This corresponds to part (A) in Figure 2 (b). The rest proof is the same as described in Section 3.1. Hence we complete the proof. $\qquad\square$

## A.5 PROOF OF LEMMA 3.3

The proof is essentially follows Altschuler & Talwar (2022; 2023). We include the proof for completeness.

*Proof.* Let $\nu$ be a probability distribution certifies $D_\alpha^{(z)}(\mu||\mu')$. That is, $D_\alpha(\nu||\mu') = D_\alpha^{(z)}(\mu||\mu')$ and $W_\infty(\nu, \mu) \le z$. Since $\phi$ is $c$-Lipschitz, we have

$$W_\infty(\phi\sharp\nu, \phi\sharp\mu) \le cW_\infty(\nu, \mu) \le c \cdot z. \tag{76}$$

It implies that $\phi\sharp\nu$ is a feasible solution for $D_\alpha^{(cz)}(\phi\sharp\mu||\phi\sharp\mu')$. Hence we have

$$D_\alpha^{(cz)}(\phi\sharp\mu||\phi\sharp\mu') \le D_\alpha(\phi\sharp\nu||\phi\sharp\mu') \overset{(a)}{\le} D_\alpha(\nu||\mu') \overset{(b)}{=} D_\alpha^{(z)}(\mu||\mu'), \tag{77}$$

where $(a)$ is due to data processing inequality of Rényi divergence and $(b)$ is due to our definition of $\nu$. Together we complete the proof. $\qquad\square$

## A.6 PROOF OF LEMMA 3.4

The proof is essentially a simple application of Lemma 3.6 of Hardt et al. (2016) or Lemma 2.2 of Altschuler & Talwar (2023). We denote $\psi(x) = x - \eta\Pi_{B_K}\nabla\ell(x)$.

When $\ell$ is $L$-smooth only, we have for any $x, y \in \mathbb{R}^d$:

$$\|\psi(x) - \psi(y)\| \le \|x - y\| + \eta\|\Pi_{B_K}\nabla\ell(x) - \Pi_{B_K}\nabla\ell(y)\| \tag{78}$$

$$\le \|x - y\| + \eta\|\nabla\ell(x) - \nabla\ell(y)\| \le \|x - y\| + \eta L\|x - y\| = (1 + \eta L)\|x - y\|. \tag{79}$$

Here we use triangle inequality, the projection operation is 1-Lipschitz and $L$-smoothness sequentially. Thus we complete the proof for the smooth case.

When $\ell$ is further $K$-Lipschitz, note that we always have $\Pi_{B_K}\nabla\ell(x) = \nabla\ell(x)$ since $\|\nabla\ell(x)\| \le K$. Thus we can drop the projection operator directly. Then the convex and $m$-strongly convex cases directly follow Lemma 3.6 of Hardt et al. (2016) or Lemma 2.2 of Altschuler & Talwar (2023). Together we complete the proof.

## A.7 PROOF OF LEMMA 3.5

This is a special case of the proof of Lemma A.2. A similar proof also appears in Chien et al. (2024b). We nevertheless prove this special case again for the readers.

Recall that by definition,

$$W_\infty(W_t, W_t') = \inf_{\gamma \in \Gamma} \operatorname*{ess\,sup}_{(X,Y) \sim \gamma} \|X - Y\|. \tag{80}$$

It means that we may choose a specific coupling between $W_t, W_t'$ to serve as an upper bound. We choose the naive coupling $G_t = G_t'$. Under this coupling, the only randomness is the initialization $W_0$. Let us denote the index that is differ between $\mathcal{D}, \mathcal{D}'$ to be $i^\star$. Then we have

$$\|W_t - W_t'\| = \|W_{t-1} - W_{t-1}' - \frac{\eta}{n} \sum_{i \in n} (\Pi_{B_k}[\nabla \ell_i(W_{t-1})] - \Pi_{B_k}[\nabla \ell_i'(W_{t-1}')])\| \tag{81}$$

$$\overset{(a)}{\leq} \frac{1}{n} \sum_{i \in [n] \setminus \{i^\star\}} \|W_{t-1} - W_{t-1}' - \eta(\Pi_{B_k}[\nabla \ell_i(W_{t-1})] - \Pi_{B_k}[\nabla \ell_i(W_{t-1}')])\| \tag{82}$$

$$+ \frac{1}{n} \|W_{t-1} - W_{t-1}' - \eta(\Pi_{B_k}[\nabla \ell_{i^\star}(W_{t-1})] - \Pi_{B_k}[\nabla \ell_{i^\star}'(W_{t-1}')])\|, \tag{83}$$

where (a) is due to the fact that $\ell_i = \ell_i'$ for all $i \neq i^\star$. For the first term, notice that the gradient update map is identical. By the assumption that the map is $c$-Lipschitz, we have

$$\frac{1}{n} \sum_{i \in [n] \setminus \{i^\star\}} \|W_{t-1} - W_{t-1}' - \eta(\Pi_{B_k}[\nabla \ell_i(W_{t-1})] - \Pi_{B_k}[\nabla \ell_i(W_{t-1}')])\| \tag{84}$$

$$\leq \frac{1}{n} \sum_{i \in [n] \setminus \{i^\star\}} c\|W_{t-1} - W_{t-1}'\| \leq c\frac{n-1}{n} \|W_{t-1} - W_{t-1}'\|. \tag{85}$$

For the second term, we may further bound it with triangle inequality.

$$\frac{1}{n} \|W_{t-1} - W_{t-1}' - \eta(\Pi_{B_k}[\nabla \ell_{i^\star}(W_{t-1})] - \Pi_{B_k}[\nabla \ell_{i^\star}'(W_{t-1}')])\| \tag{86}$$

$$\leq \frac{1}{n} \|W_{t-1} - W_{t-1}' - \eta(\Pi_{B_k}[\nabla \ell_{i^\star}(W_{t-1})] - \Pi_{B_k}[\nabla \ell_{i^\star}(W_{t-1}')])\| \tag{87}$$

$$+ \eta\frac{1}{n} \|\Pi_{B_k}[\nabla \ell_{i^\star}(W_{t-1}')] - \Pi_{B_k}[\nabla \ell_{i^\star}'(W_{t-1}')]\| \tag{88}$$

$$\leq c\frac{1}{n} \|W_{t-1} - W_{t-1}'\| + \frac{2\eta K}{n}. \tag{89}$$

Combining the two parts, we arrive the bound

$$\operatorname*{ess\,sup}_{W_0} \|W_t - W_t'\| \leq \operatorname*{ess\,sup}_{W_0} c\|W_{t-1} - W_{t-1}'\| + \frac{2\eta K}{n}. \tag{90}$$

Notice that choosing $D_t = \operatorname{ess\,sup}_{W_0} \|W_t - W_t'\|$ and $D_{t-1} = \operatorname{ess\,sup}_{W_0} c\|W_{t-1} - W_{t-1}'\|$ will give the desired recurssive relation. Also note that by definition $W_t, W_t'$ both start with the same initialization $W_0$, and thus $D_0 = 0$. Finally, notice that we always have a trivial bound $W_\infty(W_t, W_t') \leq D$ do to the projection $\Pi_\mathcal{K}$ where $D$ is the diameter of $\mathcal{K}$. Together we complete the proof.

**Comparison to Lemma 3.3 in Chien et al. (2024b).** Our Lemma 3.5 and Lemma 3.3 in Chien et al. (2024b) coincide in the strongly convex setting and have the similar spirit. However, for the general smooth case that $c > 1$, our bound can be tighter. Note that Lemma 3.3 of Chien et al. 2024 can be viewed as always leveraging the first term in the minimum of equation (7) and writing out the recursion. Indeed, when $c < 1$ the minimum will always be the first term. However, for the general case $c > 1$, it is possible that the minimum will be the second term. This happens whenever $(c-1)D_{t-1} > 2\eta K(1 - 1/n)$.

A.8 PROOF OF LEMMA 3.8

*Proof.* Let $\nu$ be a probability distribution certifies $D_\alpha^{(z)}(\mu||\mu')$. That is, $D_\alpha(\nu||\mu') = D_\alpha^{(z)}(\mu||\mu')$ and $W_\infty(\nu,\mu) \leq z$. By definition, we know that there exist a coupling $\gamma^\star \in \Gamma(\nu,\mu)$ such that

$$\operatorname*{ess\,sup}_{(X,Y)\sim\gamma^\star} \|X - Y\| \leq z. \tag{91}$$

Now, let us consider this specific coupling $\gamma^\star$ and $(X,Y) \sim \gamma^\star$. We have

$$W_\infty((I+\phi)\sharp\nu, (I+\phi)\sharp\mu) \tag{92}$$

$$\stackrel{(a)}{\leq} \operatorname*{ess\,sup}_{(X,Y)\sim\gamma^\star} \|X + \phi(X) - (Y + \phi(Y))\| \tag{93}$$

$$\leq \operatorname*{ess\,sup}_{(X,Y)\sim\gamma^\star} \|X - Y\| + \|\phi(X) - \phi(Y)\| \tag{94}$$

$$\stackrel{(b)}{\leq} \operatorname*{ess\,sup}_{(X,Y)\sim\gamma^\star} \|X - Y\| + L\|X - Y\|^\lambda \tag{95}$$

$$\leq \operatorname*{ess\,sup}_{(X,Y)\sim\gamma^\star} \|X - Y\| + \operatorname*{ess\,sup}_{(X,Y)\sim\gamma^\star} L\|X - Y\|^\lambda \tag{96}$$

$$\leq \operatorname*{ess\,sup}_{(X,Y)\sim\gamma^\star} \|X - Y\| + L(\operatorname*{ess\,sup}_{(X,Y)\sim\gamma^\star} \|X - Y\|)^\lambda \tag{97}$$

$$\stackrel{(c)}{\leq} z + Lz^\lambda, \tag{98}$$

where $(a)$ is due to the fact that any coupling $\gamma \in \Gamma$ will give an upper bound to the infimum over $\Gamma$, $(b)$ is due to the assumption that $\phi$ is $(L,\lambda)$-Hölder continuous, $(c)$ is due to our construction (91).

As a result, we know that when $W_\infty((I+\phi)\sharp\nu, (I+\phi)\sharp\mu) \leq z'$ for some $z' \geq 0$, if we can find a $z \geq 0$ such that $z' = z + Lz^\lambda$, then the construction (91) is valid. By our definition of $h$, we know that $z = h(z')$ gives such a valid choice. This implies that

$$D_\alpha^{(z')}((I+\phi)\sharp\mu||(I+\phi)\sharp\mu') \tag{99}$$

$$\stackrel{(a)}{\leq} D_\alpha((I+\phi)\sharp\nu||(I+\phi)\sharp\mu') \tag{100}$$

$$\stackrel{(b)}{\leq} D_\alpha(\nu||\mu') \tag{101}$$

$$\stackrel{(c)}{=} D_\alpha^{(z)}(\mu||\mu') \tag{102}$$

where $(a)$ is due to the fact that $\nu$ indeed satisfies $W_\infty((I+\phi)\sharp\nu, (I+\phi)\sharp\mu) \leq z'$ by setting $z = h(z')$, $(b)$ is from post-processing property of Rényi divergence (Lemma 2.3), $(c)$ is due to our construction in the beginning. By plugging in the relation $z = h(z')$ we complete the proof. $\qquad\square$

A.9 PROOF OF LEMMA 3.9

Recall that we define $h(z)$ to be the solution map of the equation $x + Lx^\lambda = z$ for all $z \geq 0$ and $x \geq 0$. When $L \geq 0, \lambda \in (0,1]$, we essentially want to show that $h$ is the inverse map of $f(x) = x + Lx^\lambda$. Observe that $\frac{d}{dx}f(x) = 1 + Lx^{\lambda-1} > 0$ for all $x \geq 0$, we know that $f(x)$ is strictly monotone increasing. Furthermore, it is not hard to see that $f$ is continuous. Then combining with facts that $f(0) = 0$ and $f(x) \to +\infty$ as $x \to +\infty$, we know that $f : \mathbb{R}_{\geq 0} \mapsto \mathbb{R}_{\geq 0}$ is bijective. As a result, $h = f^{-1}$ is bijective, $h : \mathbb{R}_{\geq 0} \mapsto \mathbb{R}_{\geq 0}$ and $h$ is strictly monotonic increasing as well. We are left to show the close-form characterization of $h$ when $\lambda = \frac{1}{2}$. In this special case, we have

$$f(x) = x + Lx^\lambda = x + L\sqrt{x}. \tag{103}$$

Let us denote $u = \sqrt{x}$, then we know that the solution of $u^2 + Lu = z$ is

$$u^\star = \frac{-L + \sqrt{L^2 + 4z}}{2}, \tag{104}$$

where we always have $u \geq 0$. As a result, the solution $h(z) = (u^\star)^2$. Thus we complete the proof.

## A.10 PROOF OF LEMMA 3.10

This is a full batch special case of Lemma A.2. We refer readers to the more general proof of Lemma A.2 and leave the proof of this special case as an exercise for readers.

## A.11 PROOF OF LEMMA A.2

Recall that by the definition of DP-SGD iterates (1) and (2), we have

$$W_\infty(W_{t+1}, W'_{t+1}) = W_\infty(\Pi_\mathcal{K}\left[W_t + \phi_t(W_t) + G'_t\right], \Pi_\mathcal{K}\left[W'_t + \phi'_t(W'_t) + G'_t\right]) \tag{105}$$

Clearly, the diameter $D$ of the projection set $\mathcal{K}$ is a natural upper bound. Also, note that $\Pi_\mathcal{K}$ is 1-Lipschitz, thus we have

$$W_\infty(W_{t+1}, W'_{t+1}) \tag{106}$$

$$\leq W_\infty(W_t + \phi_t(W_t) + G'_t, W'_t + \phi'_t(W'_t) + G'_t) \tag{107}$$

$$\overset{(a)}{\leq} \operatorname{ess\,sup} \|W_t + \phi_t(W_t) - (W'_t + \phi'_t(W'_t))\| \tag{108}$$

$$\leq \operatorname{ess\,sup} \|W_t - W'_t\| + \|\phi_t(W_t) - \phi'_t(W'_t)\| \tag{109}$$

where $(a)$ is due to the fact that we take a specific coupling $G_t = G'_t$ for all $t$. Now, let us denote the $t_e$ to be the time step that we encounter $i^\star$ (i.e., the index that $\mathcal{D}, \mathcal{D}'$ differs) at epoch $e$. Clearly, for all $t \leq t_0$, since we have not encountered $i^\star$ yet so the two adjacent processes have identical updates. Combining with the fact that $W_0 = W'_0$, we have $W_t = W'_t$. This proves the case $t \leq t_0$. For $t = t_0 + 1$, know that we have

$$W_\infty(W_{t_0+1}, W'_{t_0+1}) \leq \operatorname{ess\,sup} \|W_{t_0} - W'_{t_0}\| + \|\phi_{t_0}(W_{t_0}) - \phi'_{t_0}(W'_{t_0})\| \tag{110}$$

$$\overset{(a)}{=} \|\phi_{t_0}(W_{t_0}) - \phi'_{t_0}(W'_{t_0})\| \tag{111}$$

$$\overset{(b)}{=} \frac{\eta}{b}\|\Pi_{B_K}\nabla\ell_{i^\star}(W_{t_0}) - \Pi_{B_K}\nabla\ell'_{i^\star}(W_{t_0})\| \leq \frac{2K\eta}{b}. \tag{112}$$

where $(a)$ is due to our discuss that $W_t = W'_t$ for all $t \leq t_0$, $(b)$ is due to the fact that $\ell_i = \ell'_i$ for all $i \neq i^\star$. Thus we have proved the case $t = t_0 + 1$.

For the case $t - 1 \in \{t_e\}_{e=1}^E$, we similarly have

$$W_\infty(W_{t_e+1}, W'_{t_e+1}) \leq \operatorname{ess\,sup} \|W_{t_e} - W'_{t_e}\| + \|\phi_{t_e}(W_{t_e}) - \phi'_{t_e}(W'_{t_e})\| \tag{113}$$

$$\leq \operatorname{ess\,sup} \|W_{t_e} - W'_{t_e}\| + \frac{\eta}{b}\sum_{i\in\mathcal{B}_{t_e}}\|\Pi_{B_K}\nabla\ell_i(W_{t_e}) - \Pi_{B_K}\nabla\ell'_i(W'_{t_e})\| \tag{114}$$

$$\overset{(a)}{=} \operatorname{ess\,sup} \|W_{t_e} - W'_{t_e}\| + \frac{\eta}{b}\sum_{i\in\mathcal{B}_{t_e}\setminus\{i^\star\}}\|\Pi_{B_K}\nabla\ell_i(W_{t_e}) - \Pi_{B_K}\nabla\ell_i(W'_{t_e})\| \tag{115}$$

$$+ \frac{\eta}{b}\|\Pi_{B_K}\nabla\ell_{i^\star}(W_{t_e}) - \Pi_{B_K}\nabla\ell_{i^\star}(W'_{t_e})\| \tag{116}$$

$$\leq \operatorname{ess\,sup} \|W_{t_e} - W'_{t_e}\| + \frac{\eta}{b}\sum_{i\in\mathcal{B}_{t_e}\setminus\{i^\star\}}\|\Pi_{B_K}\nabla\ell_i(W_{t_e}) - \Pi_{B_K}\nabla\ell_i(W'_{t_e})\| \tag{117}$$

$$+ \frac{2K\eta}{b}, \tag{118}$$

where $(a)$ is due to the fact that $\ell_i = \ell'_i$ for all $i \neq i^\star$. Note that now we have two valid upper bounds. One is to utilize $\Pi_{B_K}$ directly for the rest gradient difference terms, this leads to

$$W_\infty(W_{t_e+1}, W'_{t_e+1}) \leq \operatorname{ess\,sup} \|W_{t_e} - W'_{t_e}\| + 2K\eta \overset{(a)}{\leq} D_{t_e} + 2K\eta, \tag{119}$$

where $D_t$ is the upper bound of $\operatorname{ess\,sup} \|W_t - W_t'\|$ under the coupling that $G_t = G_t'$ for all $t$. The other one is to leverage the Hölder continuous properties of $\nabla \ell_i$, which leads to

$$W_\infty(W_{t_e+1}, W_{t_e+1}') \leq \operatorname{ess\,sup} \|W_{t_e} - W_{t_e}'\| + \frac{2K\eta}{b} \tag{120}$$

$$+ \frac{\eta}{b} \sum_{i \in \mathcal{B}_{t_e} \setminus \{i^\star\}} \|\nabla \ell_i(W_{t_e}) - \nabla \ell_i(W_{t_e}')\| \tag{121}$$

$$\leq \operatorname{ess\,sup} \|W_{t_e} - W_{t_e}'\| + \frac{2K\eta}{b} + \frac{\eta(b-1)L}{b} \|W_{t_e} - W_{t_e}'\|^\lambda \tag{122}$$

$$\leq \operatorname{ess\,sup} \|W_{t_e} - W_{t_e}'\| + \frac{2K\eta}{b} + \operatorname{ess\,sup} \frac{\eta(b-1)L}{b} \|W_{t_e} - W_{t_e}'\|^\lambda \tag{123}$$

$$\overset{(a)}{\leq} D_{t_e} + \frac{\eta(b-1)L}{b} D_{t_e}^\lambda + \frac{2K\eta}{b} \tag{124}$$

$$= g(D_{t_e}; \frac{\eta(b-1)L}{b}) + \frac{2K\eta}{b}. \tag{125}$$

where $D_t$ is the upper bound of $\operatorname{ess\,sup} \|W_t - W_t'\|$ under the coupling that $G_t = G_t'$ for all $t$. By taking the minimum of the two bounds, we have proved the case $t - 1 \in \{t_e\}_{e=1}^E$. Finally, for the rest $t$ we can repeat the same analysis above, but this time we will not encounter $i^\star$ so we have $g(D_{t_e}; \eta L)$ instead. To conclude the proof, note that all our iterations for $D_t$ are derived for the specific coupling $G_t = G_t'$. Thus $D_t$ is indeed a valid upper bound for $\operatorname{ess\,sup} \|W_t - W_t'\|$. This concludes the proof.

## A.12 Proof of Corollary A.3

The proof follows by applying the joint convexity of KL divergence, which is also used in Chien et al. (2024b).

**Lemma A.5** (Lemma 4.1 in Ye & Shokri (2022)). *Let $\nu_1, \cdots, \nu_m$ and $\nu_1', \cdots, \nu_m'$ be distributions over $\mathbb{R}^d$. For any $\alpha \geq 1$ and any coefficients $p_1, \cdots, p_m \geq 0$ such that $\sum_{i=1}^m p_i = 1$, the following inequality holds.*

$$\exp((\alpha - 1)D_\alpha(\sum_{i=1}^m p_i \nu_i || \sum_{i=1}^m p_i \nu_i')) \tag{126}$$

$$\leq \sum_{i=1}^m p_i \exp((\alpha - 1)D_\alpha(\nu_i || \nu_i')). \tag{127}$$

By applying the lemma above, we can convert the $\mathcal{B}$ dependent result in Theorem A.1 to an average case bound. Thus we complete the proof.

## A.13 Numerical evaluations

**Numerical setting for Figure 1.** We choose smoothness constant $L$ and the strong convexity constant $m$ to be 1 for simplicity. We set gradient clipping norm $K = 2$, noise standard deviation $\sigma = 1.0$, domain diameter $D = 1.0$, the dataset size $n = 5$, and the step size $\eta = 0.1$.

**Numerical setting for Figure 2.** For the experiment shown in Figure 2 (a), our model is a 2-layer MLP with a hidden dimension of 64. The task is classification on the UCI Iris dataset (Fisher, 1988) as a toy example. We set the learning rate to be 0.1, training epoch 200, gradient clipping norm 1.0, and use cross-entropy loss with an additional 0.1 multiplicative factor. To estimate the Hölder constant, for each weight along the training trajectory we sampled 100 points uniformly at random from a $\ell_2$ ball of radius 1 centered at the weight. Then we compute the maximum ratio between the norm of gradient difference and weight difference of power $\lambda$, which is the estimated Hölder constant. We further repeat this process with 20 different initialization and take the maximum of the estimated Hölder constant as the final estimated for computing the privacy loss under the same setting as Figure 1.

Note that when $\lambda \to 0$, the Hölder continuous gradient condition says that the gradient difference norm is upper bounded by a constant. This in fact degenerates to the case of standard composition

theorem, since the purpose of clipped gradient norm is exactly for such condition. As a result, when $\lambda \to 0$, the bound will become identical to the Pareto frontier of the standard composition theorem and output perturbation. In addition, we also test the case $\lambda < 0.5$ and find that it will become worse whenever is too small.

We emphasize that this experiment is merely a proof of concept, and in reality, we conjecture that the loss is at least locally satisfying the Hölder continuous gradient condition. As we discussed in Section 5, one may need to explore these properties in a per-layer fashion. We also conjecture that a non-uniform Hölder continuous gradient condition involving different $\lambda$ locally is the actual case for deep neural networks.

**Computational complexity.** We optimize the proposed privacy bound as follows. For any fixed $\tau$, we optimize the privacy bound with respect to parameters $a_t, \beta_t$ by the "scipy.optimize.minimize" function in scipy library. To optimize for $\tau$, we simply use a binary search. Note that while this strategy does not guarantee to find the global optimum of the problem, we emphasize that any feasible solution gives a valid (but potentially loose) privacy bound. In our numerical settings, any point in Figure 1 (a) (i.e., smooth case) can be computed almost immediately (less than a second). For points in Figure 2 (a), the computational time is no more than a few minutes in general.

## A.14 AN EXAMPLE NON-SMOOTH FUNCTION THAT HAS HÖLDER CONTINUOUS GRADIENT

This example and the proof are from Martin R[4]. We repeat it here for completeness.

Consider $f(x) = \text{sign}(x)\frac{3}{4}|x|^{4/3}$. It is not hard to see that $G(x) = f'(x) = |x|^{1/3}$. Here we show that $G(x)$ is $(2^{2/3}, \frac{1}{3})$-Hölder continuous but not Lipshitz.

**Case 1.** $x, y \geq 0$. Without loss of generality assume $x > y \geq 0$. Then

$$x^{1/3} - y^{1/3} \leq (x - y)^{1/3} \Leftrightarrow x^{1/3} \leq (x - y)^{1/3} + y^{1/3}. \tag{128}$$

Note that for all $a, b \geq 0$, $s \in (0, 1]$, we have

$$1 = \frac{a}{a + b} + \frac{b}{a + b} \leq \left(\frac{a}{a + b}\right)^s + \left(\frac{b}{a + b}\right)^s. \tag{129}$$

This inequality is due to the fact that $z^s \geq z$ for all $s \in (0, 1]$ and $z \in [0, 1]$. Then by choosing $a = x - y$, $b = y$, we have

$$1 \leq \left(\frac{x - y}{x}\right)^{1/3} + \left(\frac{y}{x}\right)^{1/3} \Leftrightarrow x^{1/3} \leq (x - y)^{1/3} + y^{1/3} \Leftrightarrow x^{1/3} - y^{1/3} \leq 1 \cdot (x - y)^{1/3}. \tag{130}$$

Clearly $1 \leq 2^{2/3} \approx 1.587$. So our claim is true for case 1.

**Case 2.** $x, y \leq 0$. Without loss of generality assume $x < y \leq 0$. Then the same analysis from case 1 applies since $|G(x) - G(y)| = |G(-x) - G(-y)| \leq |(-x - (-y))| = |x - y|$. So our claim is true for case 2.

**Case 3.** $x, y$ are of opposite sign. Without loss of generality assume $x < 0 < y$. Then

$$|G(x) - G(y)| \leq C|x - y|^{1/3} \Leftrightarrow |x|^{1/3} + y^{1/3} \leq C(|x| + y)^{1/3} \tag{131}$$

$$\Leftrightarrow \left(\frac{|x|}{|x| + y}\right)^{1/3} + \left(\frac{y}{|x| + y}\right)^{1/3} \leq C. \tag{132}$$

Note that both fractions are less than 1, so $C$ can be chosen to be 2. A tighter estimate can be obtained by solving the maximization $u^{1/3} + (1 - u)^{1/3}$ for $u \in (0, 1)$, which gives $C = 2^{2/3}$. Together we complete the proof that $G$ is $(2^{2/3}, \frac{1}{3})$-Hölder continuous.

To show that $G$ is not Lipschitz (and thus $f$ is non-smooth), observe that for $x > 0$

$$\frac{|G(x) - G(0)|}{|x|} = x^{-\frac{2}{3}}, \tag{133}$$

which is unbounded when $x \to 0^+$. Hence $G$ is not Lipschitz continuous.

---

[4]https://math.stackexchange.com/q/3213522

### A.15 DETAILED COMPARISON TO KONG & RIBERO (2024)

In addition to the difference in the problem setting and assumptions used, we would like to add more details on the tightness aspect to compare our privacy bound.

**Comparison to output perturbation bound.** As we have shown in our Figure 1 (a), our privacy bound can be strictly and significantly better than the output perturbation bound, which is $\frac{\alpha D^2}{2\sigma^2}$, where we recall that $D$ is the domain diameter, $\sigma^2$ is the noise variance and $\alpha$ is the parameter in $\alpha$-Rényi divergence. In the meanwhile, from the main theorem (i.e., Theorem 2.2) of Kong & Ribero (2024), we can see that their bound is $\frac{\alpha}{2\sigma^2}(C_1 D + C_2)^2$ for some problem-dependent constant $C_1 > 1$ and $C_2 > 0$. Apparently, this bound is strictly worse than the output perturbation bound abovementioned. This is the first key difference between our results.

**Behavior as dataset size $n$ grows.** Intuitively, a larger dataset size $n$ should lead to smaller privacy loss for DP-SGD, since only one out of $n$ gradient term is changed between the adjacent dataset. This is apparently true for the classic composition theorem-based analysis. Note that for smooth convex losses, our result degenerates to the bound of Altschuler & Talwar (2022); Altschuler et al. (2024), which also exhibits this behavior. That is, as $n \to \infty$ the privacy loss goes to 0. In contrast, The main theorem (i.e., Theorem 2.2) of Kong & Ribero (2024) gives a bound $\frac{\alpha}{2\sigma^2}(C_1 D)^2$ in the full-batch setting when $n = b \to \infty$ for some problem-dependent constant $C_1 > 1$ independent of $n$. Apparently, this bound does not decay to 0 which is significantly different from our bound and bound in Altschuler & Talwar (2022); Altschuler et al. (2024).

