# OpenReview forum: "Convergent Privacy Loss of Noisy-SGD without Convexity and Smoothness"
_ICLR.cc/2025/Conference — ICLR 2025 Poster_

### Official Review · Reviewer_8m94 · 2024-10-27

**Soundness:** 4
**Presentation:** 3
**Contribution:** 3
**Rating:** 8
**Confidence:** 4

**Summary:**

This paper studies the problem of releasing the last iterate of noisy-SGD with Renyi DP guarantees. It presents:

- Bounds on the Renyi DP guarantee of order $\alpha$ for this problem that are independent of the number of iterations for non-convex, non-smooth functions that have a Holder continuous gradient.
- Tighter bound for strongly convex, smooth loss functions.
- These bounds hold with the following batching strategies: Full batch, sampling without replacement, and shuffled cyclic.

**Strengths:**

**New Bounds and Relevance:** The paper studies a crucial problem: finding tight privacy bounds for solutions of non-convex optimization problems. Tight privacy accounting is essential for practical implementations of DP-SGD, particularly in sensitive domains like healthcare. When assumptions about a privacy guarantee are not met (e.g. convexity), the guarantee is inapplicable, forcing the use of  overly conservative bounds that can hinder accuracy.

**Novel assumption:** The authors introduce an assumption that allows them to reuse previous techniques (shifted divergence) under a milder condition (holder continuous gradient).

**Clarity:** The techniques are clearly explained  for the full batch setting. While I did not check the math exactly, the proofs do follow logically.

**Improved bounds:** The authors provide examples where their bound improves over prior work, even in the already studies (strongly) convex setting.

**Broader impact:** The results are valuable for the broader ICLR community by enabling practical implementation of ML with sensitive data. The paper presents interesting techniques (Lemma 3.7) and assumptions (reduction of smoothness to Holder continuous gradient) in a clear and insightful way.

**Weaknesses:**

Below I state some potential weaknesses of the paper and point to specific questions in the next question that would allow me to understand better.

1. The authors present a contrived loss function that meets the Holder assumption to demonstrate it is not a vacuous  relaxation from the convex to the non-convex setting. While this function meets the assumption, it questions the applicability of their results in real world settings. (Q1)

2. Theorems 3.1, 3.6, 3.11. Present bounds parameterized by optimal values of $\tau$ and a sequence $\beta$ making it hard to parse and consequently unintuitive. It becomes hard to understand practical implications and compare to prior work, and the claims about the burning period are hard to verify. (Q2)


3. The originality of Lemma 3.5 is questionable since a similar result was introduced before by Chien et al (2024b). While the authors acknowledge this prior work, they argue that it applies to a different context, machine unlearning. However, this is a misleading distinction because machine unlearning literature also tracks unlearning with a DP definition. (Q6)

4. Minor comments:
- For a general reader it is not clear from the paragraph in the introduction why Output perturbation is not a good option. (It becomes more clear in figure 1 but I would suggest having it in the main body).

- The first sentence presents noisy SGD and DP-SGD as equivalent, which is not true, in general. Noisy SGD typically refers to SGD where gradients are noisy, but there is no expectation about privacy guarantees. DP-SGD, should provide DP guarantees which might involve more steps than Noisy-SGD, e.g.,  clipping.

- Line 96, K having diameter D would suggest the set is bounded but it does not matter for the bounds, D could be infinite and the bound still holds. I would clarify this since it makes the paper stronger :)

- I suggest proofreading for typos. A few are:
    - Line 106: “differing with one point.” differing by?

    - Fig 2 caption, line 125: “It done by constructing a coupling (G_t, G_t’), resulting a coupled process….”

    - Theorem 3.11 statement “Then the DP-SGD update (1) under without replacement…”

    - Missing parenthesis in Eq. 19.

** Update **
See discussion below addressing these weaknesses.

**Questions:**

While I believe that the topic of the paper is significant, (1) the originality of ideas is not clear to me and (2) the applicability of the results it introduces could invalidate the progress I believe the paper makes in the area. The following questions are mostly to clarify this.



**Q1** Is there a practical setting where one would use the function exhibiting Holder continuity ? Or is there another function used in a practical setting that meets this assumption?

**Q2** How tractable are the problems in theorem statements 3.1, 3.6, 3.11? I assume it is easy since the plots for different settings are provided but I would suggest:
- Can the authors clarify how the bounds are calculated for the provided plots?
- Provide simplified versions or corollaries of these theorems that give more intuitive bounds? even if they are slightly looser.
- Provide concrete examples comparing these bounds to prior work for specific parameter settings?

**Q3** From Figure 1 (b) one can assume there is a regime where the presented bounds are better than [AT22]. Is there a closed characterization for this regime? And can we compare with the convergence rate to see if this number of iterations is sufficient? This would prove the presented bounds are more practical than previous ones.

**Q4** The $\hat{L}$-Lipschitz constant estimated in Figure 2 is necessary to calibrate the noise value in practical deployment of DP-SGD. After reading the methodology in the appendix it is still unclear to me how it is estimated and if this estimate is an upper bound on the real constant. Can the authors comment on how to guarantee that $\hat{L}$ is an upper bound and that this approach would not break privacy guarantees?

**Q5** In Fig 2. a) what happens as $\lambda \to  0$? The paper only shows values for $\lambda>0.5$…

**Q6** What are the differences between the introduced Lemma 3.5 and the lemma from Chien et al.?

**Q7** I found the following related works that have related topics, can the authors comment on them and how they relate to their work?
1. [https://arxiv.org/abs/2407.06496](https://arxiv.org/abs/2407.06496)
2. [https://arxiv.org/abs/2407.05237](https://arxiv.org/abs/2407.05237)

---

> ### Author Response · Authors · 2024-11-20
>
> W1, Q1: ``The practical setting of Holder continuous gradient.``
>
> Thanks for the great comment. We would like to first emphasize that the condition of Holder continuous gradient is a generalization of the commonly studied smoothness assumption. We agree that there is currently no *popular* practical setting for the Holder continuous gradient setting that we can think of. Nevertheless, the theoretical optimization community has studied problems with such conditions for a while [ref1-3]. Also, stable positional encoding in the graph learning literature is shown to be Holder continuous [ref 7], albeit it is not the same as the Holder continuous gradient condition. It is interesting to investigate what structural property we can leverage for a popular class of neural networks, which is an important future direction. We believe our work serves as an important first step toward this goal.
>
> W2, Q2: ``How to compute the bound in the main results? Is it possible to have a closed-form even for a loose bound?``
>
> This is a great question. We first answer the question of computation and then comment on the close-form of our results.
> The computation of the optimization in our main theorems is simple. First note that any (sub-optimal) feasible solutions are valid bounds. As a result, we first fix a choice of $\tau$ and then utilize the ``minimize`` function of scipy.optimize in scipy library to solve $\beta$. To optimize for $\tau$, we simply use a binary search. For the smooth setting, this routine is very efficient where Figure 1 (a) can be computed in a few seconds. For the Holder continuous gradient case, computing one point in Figure 2 (a) takes longer but still within a few minutes.
> Regarding the close-form of our results, note that we can indeed obtain close-form bounds for special cases. For example, if the loss is smooth and convex, we know that the gradient update would be 1-Lipschitz (i.e., $c=1$). This allows us to further simplify the bounds and solve the optimization directly. In fact, this degenerates to the results of Altschuler & Talwar (2022). For general smooth loss cases, we have $c\neq 1$ but a close-form solution is still possible. Unfortunately, the expression is complicated so we do not include it in our manuscript. For non-smooth losses with Holder continuous gradient, the close-form bound is unfortunately intractable. Nevertheless, we agree with reviewer 8m94 that it is possible to have a looser but close-form bound. We will investigate this further in our future work, and we really appreciate the insightful comment.
> Regarding the comparison to prior works, we believe Figure 1 (b) serves exactly this purpose. Specifically, all bounds in Figure 1 (b) are compared under the same parameter setting.
>
> Q3: ``Comparison with [AT22]``
>
> Thanks for the great comment. Note that the result of Altschuler & Talwar (2022) can be roughly understood as our Theorem 3.1 but without the forward Wasserstein tracking part. This means that the minimum term in equation (3) is replaced by $D^2$. Roughly speaking, whenever the minimum term in our equation (3) not happening at $D$ (i.e., $2\eta K/n (1-c^\tau)/(1-c)< D$), our bound will be better. Notably, the original result of Altschuler & Talwar (2022) is even worse since they simply choose a specific sub-optimal shift allocation of $a_t$ (i.e., $a_t = 0$ for all $t<T$ and $a_{T-1} = c^{T-\tau}D$. For simplicity, we have neglected this effect and adopted the optimal shift allocation for them in the discussion above and Figure 1 (a).
>
> Q4: ``Question about Figure 2 (a)``
>
> Sorry for the confusion. The experiment in Figure 2 (a) only tries to convey the message that a smaller order of Hölder continuous $\lambda$ can potentially lead to a much smaller constant $L$, which in combination gives a better privacy bound. This is in the same spirit as [ref 4], which also assumes the local smoothness assumption and estimates the corresponding constant empirically. That being said, we are not asserting that the neural network indeed has Holder continuous gradients and our estimated constant serves as a valid upper bound. We will try to make it clear in our revision. Please let us know if you have further questions on how this experiment is done. We are happy to answer it as soon as possible.

---

> ### Author Response · Authors · 2024-11-20
>
> Q5: ``In Figure 2 (a), what happen when $\lambda \rightarrow 0$?``
>
> This is a great question! Note that when $\lambda \rightarrow 0$, the Holder continuous gradient condition says that the gradient difference norm is upper bounded by a constant. This in fact degenerates to the case of standard composition theorem, since the purpose of clipped gradient norm is exactly for such condition. As a result, when $\lambda \rightarrow 0$, the bound will become identical to the Pareto frontier of the standard composition theorem and output perturbation. In our experiment, we did test the case $\lambda < 0.5$ and find that it will become worse whenever $\lambda$ is too small.
>
> W3, Q6: ``Comparison of Lemma 3.5 and the lemma in Chien et al. 2024.``
>
> Thanks for the great comment. Indeed, Lemma 3.5 and Lemma 3.3 in Chien et al. 2024 coincide in the strongly convex setting. However, for the general smooth case that $c>1$, our bound can be tighter. Note that Lemma 3.3 of Chien et al. 2024 can be viewed as always leveraging the first term in the minimum of equation (7) and writing out the recursion. Indeed, when $c<1$ the minimum will always be the first term. However, for the general case $c>1$, it is possible that the minimum will be the second term. This happens whenever $(c-1)D_{t-1} > 2\eta K(1-1/n)$. Nevertheless, we agree with the comment of reviewer 8m94 and we will give more credit to Chien et al. 2024 when we mentioned Lemma 3.5.
>
> Q7: ``Relations to recent related works.``
>
> Thanks for the references. These two works are indeed relevant to ours and we will include them in the related works section in our revision. For the work [ref 5], the author shows that not all non-convex losses can be benefitted from hidden-state in privacy. This does not contradict our results, as we show that some “continuity” in gradient (i.e., Holder continuous gradient) is needed for the reduction lemma type of result. The work [ref 6] studies a different condition where they have a parameter to control the degree of convexity. As mentioned by the author in the public comment above, our results are tighter by roughly a factor of 8 and we study the Holder continuous gradient condition that relaxes the smoothness assumption. We believe both of our works have their own contributions and complement each other.
>
> ## References
>
> [ref 1] On the quality of first-order approximation of functions with Hölder continuous gradient, Berger et al., JOTA 2020.
>
> [ref 2] Universal gradient methods for convex optimization problems, Yurri Nesterov, Mathematical Programming 2015.
>
> [ref 3] Bundle-level type methods uniformly optimal for smooth and nonsmooth convex optimization, Guanghui Lan, Mathematical Programming 2015.
>
> [ref 4] Why gradient clipping accelerates training: A theoretical justification for adaptivity, Zhang et al., ICLR 2020.
>
> [ref 5] It’s Our Loss: No Privacy Amplification for Hidden State DP-SGD With Non-Convex Loss. Meenatchi Sundaram Muthu Selva Annamalai, AISec 2024.
>
> [ref 6] Privacy of the last iterate in cyclically-sampled DP-SGD on nonconvex composite losses, Kong et al., Arxiv 2024.
>
> [ref 7] On the Stability of Expressive Positional Encodings for Graphs, Huang et al., ICLR 2024.

---

> > ### Comment · Reviewer_8m94 · 2024-11-26
> > **Thanks!**
> >
> > Thanks for the detailed and clear response to all my questions.
> >
> > **W1, Q1**
> >
> > > there is currently no popular practical setting for the Holder continuous gradient [...] It is interesting to investigate what structural property we can leverage for a popular class of neural networks, which is an important future direction. We believe our work serves as an important first step toward this goal.
> >
> > Agreed, I think that W1 still holds but I agree with the authors that studying problems beyond convexity and characterizations that allow for meaningful derivations is an open problem within the optimization community.
> >
> > **Q2 & Q3**
> >
> > Thanks, this is more clear now. However,
> > - You need access to the constants $L, \lambda$ to solve the optimization problem which is typically unavailable and,
> > - Does it also hold in the mini-batch case? the manuscript focuses mostly on the full-batch setting.
> >
> > **Q4 & Q5**
> >
> > Thanks for being transparent, however none of these has  been addressed in the manuscript yet and I think clarifying it is crucial to (1) avoid future faulty implementations and (2) parameter regimes of improvement.
> >
> >
> > **Q6 & Q7**
> >
> > Please make sure to address this in the manuscript.

---

> > > ### Author Response · Authors · 2024-11-26
> > >
> > > Thanks for your feedback! We will definitely incorporate our discussions into the revision. Below we further clarify the follow-up questions regarding Q2 & Q3.
> > >
> > > > You need access to the constants $L,\lambda$ to solve the optimization problem which is typically unavailable.
> > >
> > > Yes, reviewer 8m94 is right. There is indeed a theory-practice gap regarding this. Nevertheless, we would like to highlight that the same issue exists for all prior works in this direction. For instance, in Theorem 5.1 of [AT22] (arxiv version), they also require to know the smooth and strongly convex constant in order to know the important constant $c$ in their privacy bound for accounting. Similar issues exist for Ye & Shokri (2022). As a result, we agree with the point raised by reviewer 8m94 but that issue persists in all prior works, which is a theory-practice gap and an interesting future direction.
> > >
> > > > Does it also hold in the mini-batch case?
> > >
> > > Yes, all our results and statements have their mini-batch counterparts, where we defer all the details in Appendix A.2 and A.3 due to their complexity. We guess the reviewer 8m94 is familiar with the work [AT22] so we would suggest the setting studied in Appendix A.3 since it is the same mini-batch setting as in [AT22]. Indeed, the only difference under this setting is that Part (A) in
> > >  our Figure 2 (b), or equivalently the first term in equation (10), is now handled by Sampled Gaussian Mechanism (i.e., the $S_\alpha$ function) described in Appendix A.3 or Lemma 2.11 in [AT22]. Apparently, this change would not affect the analysis in the second term in equation (10). So our argument in the response to Q3 still holds. Still, note that the resulting optimal $\beta, \tau$ will change and thus the final privacy bound (after solving the optimization) will be different than the full batch case.
> > >
> > > Please let us know if there are further questions. We really appreciate your thoughtful comments that lead to a fruitful discussion.

---

> > > > ### Comment · Reviewer_8m94 · 2024-11-27
> > > >
> > > > Thanks for addressing my concerns. In summary, I believe this paper represents *a step* towards realistic accounting for DP-SGD by upper bounding the privacy loss of the last iterate for *a specific class* of non-convex objective functions. The paper is well organized and clearly written. It introduces novel theoretical techniques which I believe are valuable for the community to keep working on this problem and bridging the gap between theoretical and practical accounting.
> > > >
> > > > The authors addressed my concerns regarding:
> > > >
> > > > - interpretability of bounds,
> > > > - limitations of assumptions (Holder gradient), knowledge of constants,
> > > > - differences with previous and concurrent work and how these works are leveraged, different, and/or complementary. (Particularly Chien et al. 2024. and Kong et al. 2024)
> > > >
> > > > In addition, other comments point out the following:
> > > > - Comparative analysis: the authors should expand a bit on this to acknowledge previous results. Weiwei Kong points out that  their paper "*resolves the issues of gradient clipping*", which I believe this paper does not address.
> > > > - Closed form bounds: While the authors explained how to easily obtain these for given parameters, adding more intuitive explanations in the manuscript would be more helpful for readers than the optimization form.
> > > >
> > > > If the authors commit to incorporating these edits transparently in their manuscript, I believe this is a good paper and I am willing to increase my score.

---

> > > > > ### Author Response · Authors · 2024-11-27
> > > > >
> > > > > Thanks for your detailed summary! Yes, we will definitely include all the fruitful discussions during the rebuttal in our revision. Our manuscript is only 9 pages now, so we have one more additional page to incorporate our discussions. It is currently not possible to update the PDF though and we also want to take time to add these changes properly.

---

### Official Review · Reviewer_tQqF · 2024-10-28

**Soundness:** 3
**Presentation:** 3
**Contribution:** 2
**Rating:** 6
**Confidence:** 3

**Summary:**

This paper investigates the convergent privacy loss of noisy SGD within a convex bounded domain. Applying standard composition results indicates that the final privacy loss after $T$ iterations can scale with $\sqrt{T}$. However, if only the last iterate is revealed, two recent works (Ye & Shokri, 2022; Altschuler & Talwar, 2022) demonstrate that running noisy SGD on smooth and (strongly) convex functions can result in convergent privacy loss after a burn-in phase. This paper enhances the shifted divergence analysis, presenting a strictly better privacy bound under the same assumptions, and also provides convergent privacy bounds for non-convex functions with Hölder continuous gradients.

**Strengths:**

This paper is well-written and well-structured, clearly discussing prior works and their methodologies. The topic of convergent privacy loss in noisy SGD is significant, and this work makes meaningful progress in addressing it. Additionally, the results concerning non-convex functions, which can outperform output perturbation, are particularly intriguing.

**Weaknesses:**

The novelty and improvements presented are somewhat limited. The Forward Wasserstein distance tracking lemma appears straightforward, and the primary enhancement in the shifted divergence analysis seems to lie in identifying a better allocation of shifts. Regarding the non-convex case, I am concerned that the improvements may not be substantial when compared to output perturbation in practical applications. Hence,  I would not be surprised if other reviewers lean towards rejecting the paper, given the concerns raised.

**Questions:**

Is there a typo in Equation (11)?
Do any classic neural networks exhibit Hölder continuous gradients? Why is this assumption appropriate?

---

> ### Author Response · Authors · 2024-11-20
>
> Q1: ``Typo in equation (11)``
>
> Thanks for the catch! Yes, there is a typo, where there should not be an $\alpha$ in the second term in front of $a_t^2$. No other results are affected though and we will correct it in our revision.
>
> Q2: ``Why Hölder continuous gradients assumption is appropriate?``
>
> We apologize for the confusion. It is unclear if neural networks indeed have Hölder continuous gradients in general. The experiment in Figure 2 (a) only tries to convey the message that a smaller order of Hölder continuous $\lambda$ can potentially lead to a much smaller constant $L$, which in combination gives a better privacy bound. This is in the same spirit as [ref 1], which also assumes the local smoothness assumption and estimates the corresponding constant empirically. It is interesting to investigate what structural property we can leverage for a popular class of neural networks, which is an important future direction. Nevertheless, the Hölder continuous gradient assumption is indeed a generalization of the commonly used smoothness assumption. As a result, we believe our contributions here are still meaningful.
>
> ## Reference
>
>  [ref 1] Why gradient clipping accelerates training: A theoretical justification for adaptivity, Zhang et al., ICLR 2020.

---

> > ### Comment · Reviewer_tQqF · 2024-11-26
> >
> > Thanks for your reply. I will keep my score.

---

> > > ### Author Response · Authors · 2024-11-27
> > >
> > > Thanks for the acknowledgement!

---

### Official Review · Reviewer_WAgt · 2024-11-02

**Soundness:** 4
**Presentation:** 3
**Contribution:** 3
**Rating:** 5
**Confidence:** 3

**Summary:**

This paper investigates the differential privacy (DP) guarantees of Noisy Stochastic Gradient Descent (Noisy-SGD) without relying on traditional assumptions of convexity or smoothness in the loss function, thereby addressing significant limitations in prior research. Traditional DP analysis for Noisy-SGD often assumes convexity and smoothness, restricting its applicability to various real-world machine learning problems. The authors demonstrate that it is possible to achieve convergent privacy loss bounds even when these assumptions are relaxed, requiring only that the loss function has a Hölder continuous gradient. This generalization broadens the applicability of DP-SGD to a wider range of non-smooth, non-convex problems.

#### Main Contributions:
1. **Generalized Privacy Bounds**: Establishes a new privacy bound for Noisy-SGD that applies to non-convex, non-smooth losses under the less restrictive condition of Hölder continuity.
2. **Improved Privacy Bounds for Smooth Convex Losses**: Refines analysis techniques involving forward Wasserstein distance tracking and optimal shift allocation, resulting in improved privacy bounds for cases with smooth, strongly convex losses.

By advancing hidden-state analysis, this work deepens the understanding of DP in machine learning models and introduces a more flexible framework for balancing privacy and utility in Noisy-SGD, making it more suitable for deep learning applications.

**Strengths:**

The strengths of this paper lie in its innovative approach to privacy loss analysis in Noisy-SGD. Unlike prior work that requires assumptions of smoothness and convexity, this paper successfully establishes privacy bounds under more general conditions, making it applicable to a broader range of non-smooth, non-convex problems. Key strengths include:

1. **Generalization with Hölder Continuity**: The paper introduces the concept of Hölder continuous gradients, allowing the privacy analysis to hold even without the smoothness assumption. This broadens the applicability of Noisy-SGD to various challenging settings.

2. **Tighter Privacy Bounds**: For cases involving smooth and strongly convex losses, the paper presents a stricter privacy bound than prior results, offering enhanced privacy-utility trade-offs.

3. **Refined Analytical Techniques**: By optimizing privacy loss bounds through careful shift allocation and forward Wasserstein distance tracking, the analysis is robust and offers concrete improvements over existing methods.

Overall, this work contributes a significant advancement in privacy-preserving machine learning by providing a more flexible and precise analysis framework that can benefit a wide range of applications.

**Weaknesses:**

1. **Unclear Structure**: The structure of the paper lacks clarity. The main text includes numerous proofs, while some theorem statements are placed in the appendix, which disrupts the flow and clarity of the overall structure.

2. **Figure Quality**: Figure 2(a) appears somewhat rough and could benefit from refinement to improve visual quality.

3. **Typo in Lemma 2.3**: There is a typo in Lemma 2.3, where the function notation shifts from \( f \) to \( h \), which should be consistent.

4. **Overly Complex Parameters**: The large number of parameters results in overly complex expressions, making it difficult to clearly interpret and compare the results.

5. **Lack of Comparative Analysis**: The paper lacks a comparison with prior results, as well as with the non-private version of the results, which would provide a better context and understanding of its contributions.

**Questions:**

1. **Comparison with Non-Private and Smooth/Convex Cases**: Could you provide more detail on any existing comparison with non-private results? Additionally, how do these results compare specifically to the smooth and convex cases?

2. **Simplified Upper Bound for Summations**: Is it possible to present a more concise upper bound for the summations in the final results to enhance readability and ease of comparison?

3. **Challenges in Removing Smoothness and Convexity**: While the paper does extend to non-smooth and non-convex scenarios based on the condition of Hölder continuity, the results appear inherently limited by this assumption. I am particularly interested in understanding the main challenges that arise when attempting to remove the smoothness and convexity requirements entirely.

4.** Lower bound and tightness**:  Is the result in the paper tight? In other words, is there a lower bound provided?

---

> ### Author Response · Authors · 2024-11-20
>
> We thank reviewer WAgt for their positive assessment of our theoretical contribution and valuable feedback. We address the proposed questions and weaknesses below.
>
> Q1: ``Comparison with non-private results``
>
> Thanks for the suggestion. Our work currently focuses on the analysis of the privacy bound of the noisy SGD algorithm. Since non-private counterparts do not have privacy bounds, we do not think it is possible to make such a comparison from the privacy aspect. Nevertheless, we conjecture the reviewer WAgt might indicate the comparison of the other aspects, such as the utility bound. For smooth strongly convex losses, the formal utility bound for noisy SGD can be adapted from Chourasia et al. 2021. We will add it to our revision.
>
> Q2: ``Simplified privacy bound for the main results``
>
> We agree. We have tried hard to further simplify it but unfortunately, we could not find a close-form solution for general cases. As we mentioned in response to Q2 of reviewer rdb2, there is indeed a special case (i.e., smooth convex loss) where the closed-form bound is available. We feel that it would be hard to get a tight close-form solution in general but it is possible to have a close-form sub-optimal bound. On the other hand, we want to mention that many current state-of-the-art privacy accounting also do not have close-form bounds and rely on numerical computations, such as the seminal work of Gopi et al. 2021. Making the privacy bound as tight as possible, including the constants, is of significant importance in DP practice. This is also the reason why we keep the tightest possible computable privacy bound as our main theorems.
>
> Q3: ``Challenges in removing smoothness and convexity assumptions.``
>
> This is a great question. Firstly, although we do not prove it, if there is no assumption on the loss function, there will not be any ***reduction lemma*** pertaining to the gradient update in the worst-case scenario. As a result, we need some ***continuity*** of the gradient update $\psi$ for a non-trivial privacy bound compared to output perturbation asymptotically. In this work, we choose Holder continuous gradient as such assumption but other assumptions of similar spirit can be adopted.
>
> Regarding the difficulty of removing the convexity assumption, the key is to observe that in Theorem 3.1, even for the non-convex case so that $c>1$, we still have a non-trivial privacy bound. Note that when $c>1$, the gradient update map $\psi$ is not a contraction mapping anymore. For relaxing the smoothness assumption, the key is to prove the ***reduction lemma*** pertaining to $\psi$ without smoothness as in our Lemma 3.7.  Please also check lines 140-154 for a more thorough discussion.
>
> Q4: ``Lower bound aspect``
>
> This is a great question. Note that in the special case where the loss is smooth and convex, our results degenerate to the bound in Altschuler & Talwar (2022), which is proven to be tight up to a constant. We conjecture a similar worst-case construction to Altschuler & Talwar (2022) can be adapted to our general settings and prove the corresponding tight lower bound, which we left as a future work.
>
> W1: ``Typos and Figure quality``
>
> We appreciate the catch and suggestions. We will correct and improve them in our revision.
>
> W2: ``Unclear structure and clarity``
>
> As mentioned by the reviewer WAgt, we have a lot of results for different cases and scenarios. Due to the space limit, we decide to only put the full “sketch of proof” of the simplest case in the main text. Nevertheless, we believe it captures most of the key ideas of the proofs of more complex cases. We also believe it is the most accessible one for the general audience, and leave more complex and technical details in the Appendix for the experts.

---

> > ### Author Response · Authors · 2024-11-20
> >
> > W3: ``Lack of comparative analysis``
> >
> > Thanks for the comment. We have compared our result with prior works, specifically Altschuler & Talwar (2022) and Ye & Shokri (2022), in Figure 1(b). We also compare the required assumptions to these work in Table 1. Since our results are not presented in closed-form, it is difficult to compare the bound explicitly to prior works. Nevertheless, note that for the simple special case such as convex loss with proper step size, from Lemma 3.4 we know that the gradient update $\psi$ is 1-Lipschitz. In this case, a closed-form solution of Theorem 3.1 can be obtained, which is in fact the result of Altschuler & Talwar (2022).
> >
> > On the other hand, the result of Altschuler & Talwar (2022) can be roughly understood as our Theorem 3.1 but without the forward Wasserstein tracking part for the smooth and strongly convex loss. This means that the minimum term in equation (3) is replaced by $D^2$. Roughly speaking, whenever the minimum term in our equation (3) is not happening at $D$ (i.e., $2\eta K/n (1-c^\tau)/(1-c)< D$), our bound will be better. Notably, the original result of Altschuler & Talwar (2022) is even worse since they simply choose a specific sub-optimal shift allocation of $a_t$ (i.e., $a_t = 0$ for all $t<T$ and $a_{T-1} = c^{T-\tau}D$. For simplicity, we have neglected this effect and adopted the optimal shift allocation for them in the discussion above and Figure 1 (a). We will try to add this explanation in our revision.

---

> > > ### Author Response · Authors · 2024-11-30
> > > **Any unclarified concerns?**
> > >
> > > Dear reviewer WAgt,
> > >
> > > Thank you for your review and we really appreciate your help in improving our manuscript. We have noticed that reviewer WAgt gave positive assessments (Soundness: 4: excellent, Presentation: 3: good, Contribution: 3: good) but an overall borderline reject rating of 5. We wonder if this is either a mistake or if reviewer WAgt indeed has particular concerns about our work that are not clarified yet. Please do not hesitate to let us know as the rebuttal period is ending soon.

---

### Official Review · Reviewer_rdb2 · 2024-11-04

**Soundness:** 4
**Presentation:** 4
**Contribution:** 4
**Rating:** 8
**Confidence:** 3

**Summary:**

The paper refines the hidden-state privacy analysis of Noisy-SGD algorithm under various loss assumptions from strong-convex to non-convex and from Lipschitz smooth to Holder smooth. Building on the analysis of Feldman et al., 2018, and Altschuler & Talwar, 2022, the paper shows that the shift-reduction can be applied in a more effective way by noting that a better bound on the forward Wasserstein distance is possible rather than relying on the degenerate domain diameter. The paper also notes that the Lipschitz-reduction lemma of Altschuler & Talwar, 2022 can be generalized to a Holder-reduction lemma that applies to maps that satisfy the weaker condition of Holder-continuity. Applying the findings, the paper shows better DP bounds than existing work under strongly convex and smooth losses. For convex and smooth losses, their bounds matches that of Altschuler & Talwar, 2022. Additionally, for non-convex but smooth losses, the paper promises a better DP bound than the best via composition or that of output perturbation (or any Pareto-optimal combination of the two).

**Strengths:**

- The paper improves upon the state-of-the art DP bounds for Noisy-SGD on strongly-convex and smooth losses.
- The relaxation of assumption on Lipschitz-continuity to Holder-continuity is a step forward towards assumption lean convergent DP bound for Noisy-SGD.
- The paper provides DP bounds under two batch subsampling techniques that are most common.
- The ideas in the paper are well illustrated and presented well for a technical audience.
- The paper claims that under non-convex but smooth losses, their DP bound remains finite with number of iteration, unlike composition based-bounds.

**Weaknesses:**

- The main results in Theorem 3.1, 3.6 and 3.11 and Theorem aren't provided in a closed-form. This makes them hard to operationalize.
- The computational complexity of the presented bounds should be discussed.
- Full batch setting of strongly-convex and smooth case in Theorem 3.1 isn't compared with the bound in Chourasia et al., under identical assumptions.
- Utility analysis is missing. For (strongly) convex and smooth losses, the improved DP bounds can yield a better utility bounds that should be compared against the known lower-bounds [1].

[1] Bassily, Raef, Adam Smith, and Abhradeep Thakurta. "Private empirical risk minimization: Efficient algorithms and tight error bounds." 2014 IEEE 55th annual symposium on foundations of computer science. IEEE, 2014.

**Questions:**

- In equations on lines 297-305, the inequalities are under the conditioning that $Z_{1:T-1}=Z_{1:T-1}'$, right? Could the authors clarify how the dependence assumptions of Lemma 3.2 and 3.3 holds for applying the (b) inequality?
- Can the bounds presented be computed exactly or we need numerical approximations to realize them?

---

> ### Author Response · Authors · 2024-11-20
>
> We thank reviewer rdb2 for their positive comments and valuable feedback. We address the proposed questions and weaknesses below.
>
> Q1:  ``Questions about inequalities in lines 297-305.``
>
> Yes, reviewer rdb2 understands it correctly. The analysis in lines 297-305 is conditioned on $Z_{\tau:T-1}=Z_{\tau:T-1}^\prime$, which is the same as in Altschuler & Talwar (2022).
>
> Regarding the details of inequality (b) and their relation to Lemma 3.2, 3.3, we first apply the shift reduction lemma (Lemma 3.2) to turn the noise $Y_{T-1}$ in the Renyi divergence to an additive term in inequality (b). Note that there is no assumption needed at this step, since by our choice that $a_t\geq 0$ and $Y_{T-1}$ is indeed Gaussian noise. Then we apply Lemma 3.3, which removes the map $\psi$ in the Renyi divergence. Note that in this step, we need the map $\psi$ to be $c$-Lipschitz, where from Lemma 3.4 we know that the gradient update $\psi$ is indeed $c$-Lipschitz with $c$ depending on the assumptions on the loss. As a result, Lemma 3.2 add $a_{T-1}$ to the shift and Lemma 3.3 scale it by $c^{-1}$. In summary, condition on the event $Z_{\tau:T-1}=Z_{\tau:T-1}^\prime$, as long as the gradient update map $\psi$ is $c$-Lipschitz the inequality (b) will hold.
>
> Q2: `` Can the bounds presented be computed exactly or do we need numerical approximations to realize them?``
>
> This is a great question. For the simple special case such as convex loss with proper step size, from Lemma 3.4 we know that the gradient update $\psi$ is 1-Lipschitz. In this case, a closed-form solution of Theorem 3.1 can be obtained, which is in fact the result of Altschuler & Talwar (2022). For general smooth loss cases, we have $c\neq 1$ but a closed-form solution is still possible. Unfortunately, the expression is complicated so we do not include it in our manuscript. For non-smooth losses with Holder continuous gradient, the close-form bound is unfortunately intractable. Nevertheless, we would like to emphasize that the optimization in both Theorem 3.1, 3.6, and 3.11 are for the purpose of ***best possible*** privacy bound. In fact, any sub-optimal feasible solution still gives a valid privacy bound (although looser), which is different from the situation of common privacy accounting for $(\epsilon,\delta)$. It is an interesting direction to further investigate close-form (suboptimal) solutions of both Theorem 3.1, 3.6, and 3.11.
>
> W1: ``Main theorems are not in close-form and hard to operationalize. ``
>
> We agree. We have tried hard to further simplify it but unfortunately, we could not find a close-form solution for general cases. As we mentioned in the response to Q2, there is indeed a special case (i.e., smooth convex loss) where the closed-form bound is available. We feel that it would be hard to get the tight close-form solution in general but it is possible to have a close-form sub-optimal bound. On the other hand, we want to mention that many current state-of-the-art privacy accounting also do not have close-form bounds and rely on numerical computations, such as the seminal work Gopi et al. 2021. Making the privacy bound as tight as possible, including the constants, is of significant importance in DP practice. This is also the reason why we keep the tightest possible computable privacy bound as our main theorems.
>
> W2: ``Computational complexity of the bound should be discussed.``
>
> Thanks for the great comment. We will add it to our revision. To get a feeling, any point in Figure 1 (a) (i.e., smooth case) can be computed almost immediately (less than a second). For points in Figure 2 (a), the computational time is no more than a few minutes in general. Compared to the training time of modern ML models this should be negligible.
>
> W3: ``Did not compare to Chourasia et al. 2021 for the smooth strongly convex case.``
>
> Note that as mentioned by Ye & Shokri (2022), specifically in their Appendix D.7, their bound is tighter than Chourasia et al. 2021 for the smooth strongly convex case. We have compared with Ye & Shokri (2022) as in Figure 1 (a) and we think it is sufficient, where we have cited Chourasia et al. 2021 in the related work section as well.
>
>
> W4: ``No utility bound``
>
> This is a great comment. Indeed, it would be great to have a delegate utility bound analysis. Unfortunately, this is hard in general to the best of our knowledge. For simpler special cases such as smooth strongly convex losses, the corresponding utility results can be adapted from Chourasia et al. 2021. We will add it to our revision and discuss the lower bound aspect [1].

---

> > ### Comment · Reviewer_rdb2 · 2024-11-25
> >
> > Thank you for the rebuttal. All your comments are well received. I maintain my score.

---

> > > ### Author Response · Authors · 2024-11-25
> > >
> > > Thank you for letting us know! We truly appreciate your thoughtful comments and your help in improving our manuscript.

---

### Public Comment · ~Weiwei_Kong1 · 2024-11-18
**Related work**

Just to add to the discussion - my group has a prior/concurrent (July 2024) preprint that follows a similar approach as the authors ([link](https://arxiv.org/abs/2407.05237)):

* Kong, W., & Ribero, M. (2024). Privacy of the last iterate in cyclically-sampled DP-SGD on nonconvex composite losses. *arXiv preprint arXiv:2407.05237*.

More specifically, our work develops closed-form Renyi differential privacy bounds using similar techniques, but tackles and resolves the issues of gradient clipping, assuming Lipschitz continuity of the objective function, bounded domains, and continuity (to the convex setting).

At the same time, I believe the submitted manuscript develops tighter Renyi differential privacy bounds, more generally assumes that the gradient of the objective function is Holder continuous, and considers the case of shuffled batches in Appendix A.2.2.

---

> ### Author Response · Authors · 2024-11-25
>
> Hi Weiwei Kong
>
> Thanks for introducing your concurrent work and a comparison. We believe both of our works have their own contributions and complement each other. In addition to the difference in the problem setting and assumptions used, we would like to add more details on the tightness aspect to compare our privacy bound. We believe this will also help reviewers and post readers understand our works' differences.
>
> ## Comparison to output perturbation bound.
>
> As we have shown in our Figure 1 (a), our privacy bound can be strictly and significantly better than the output perturbation bound, which is $\frac{\alpha D^2}{2\sigma^2}$, where $D$ is the domain diameter, $\sigma^2$ is the noise variance and $\alpha$ is the parameter in $\alpha$ Renyi divergence. In the meanwhile, from the main theorem (i.e., Theorem 4.3) of Kong, W., & Ribero, M. (2024), we can see that their bound is $\frac{\alpha}{2\sigma^2}(C_1 D + C_2)^2$ for some problem-dependent constant $C_1>1$ and $C_2>0$. Apparently, this bound is strictly worse than the output perturbation bound abovementioned. This is the first key difference between our results.
>
> ## Behavior as dataset size $n$ grows.
>
> Intuitively, a larger dataset size $n$ should lead to smaller privacy loss for DP-SGD, since only one out of $n$ gradient term is changed between the adjacent dataset. This is apparently true for the classic composition theorem-based analysis. Note that for smooth convex losses, our result degenerates to the bound of Altschuler & Talwar (2022), which also exhibits this behavior. That is, as $n\rightarrow\infty$ the privacy loss goes to $0$. In contrast, The main theorem (i.e., Theorem 4.3) of Kong, W., & Ribero, M. (2024) gives a bound $\frac{\alpha}{2\sigma^2}(C_1 D)^2$ in the full-batch setting when $n=b \rightarrow \infty$ for some problem-dependent constant $C_1>1$ independent of $n$. Apparently, this bound does not decay to $0$ which is significantly different from our bound and bound in Altschuler & Talwar (2022).
>
> Nevertheless, we emphasize that there are still scenarios where the result of Kong, W., & Ribero, M. (2024) is preferable. Like what Weiwei mentioned in their post, they can deal with clipped gradients without assuming the loss to be Lipschitz and they have a characterization of the different degrees of convexity in their bound (i.e., $C_1$). We will cite the work and add the discussion above to our revision. Really appreciate your introduction to your work!

---

### Meta-Review · Area_Chair_Cf7e · 2024-12-19

**Metareview:**

The paper got sufficient support from the reviewers (with a few of them have historically worked on similar problems). However, there were a few major concerns, a) The assumption on Holder continuous gradient is unclear how operational it is for natural optimization tasks, and b) the extension to privacy amplification by iteration is not very novel, and c) there were concerns on the comparability of the result to prior results using output perturbation.

**Additional Comments On Reviewer Discussion:**

I think the authors addressed most of the concerns, except the ones stated in the meta-review.

---

### Decision · Program_Chairs · 2025-01-22

Accept (Poster)